# Integration of daytime radiative cooling and solar heating for year-round energy saving in buildings

Xiuqiang Li[1], Bowen Sun [1], Chenxi Sui[1], Ankita Nandi [2], Haoming Fang [1], Yucan Peng[3], Gang Tan [4✉] & Po-Chun Hsu [1✉]

The heating and cooling energy consumption of buildings accounts for about 15% of national total energy consumption in the United States. In response to this challenge, many promising technologies with minimum carbon footprint have been proposed. However, most of the approaches are static and monofunctional, which can only reduce building energy consumption in certain conditions and climate zones. Here, we demonstrate a dual-mode device with electrostatically-controlled thermal contact conductance, which can achieve up to 71.6 W/m$^2$ of cooling power density and up to 643.4 W/m$^2$ of heating power density (over 93% of solar energy utilized) because of the suppression of thermal contact resistance and the engineering of surface morphology and optical property. Building energy simulation shows our dual-mode device, if widely deployed in the United States, can save 19.2% heating and cooling energy, which is 1.7 times higher than cooling-only and 2.2 times higher than heating-only approaches.

[1] Department of Mechanical Engineering and Material Science, Duke University, Durham, NC 27708, USA. [2] North Carolina School of Science and Mathematics, Durham, NC 27705, USA. [3] Department of Materials Science and Engineering, Stanford University, Stanford, CA 94305, USA. [4] Department of Civil and Architectural Engineering, University of Wyoming, Laramie, WY 82071, USA. ✉email: gtan@uwyo.edu; pochun.hsu@duke.edu

The energy consumption in buildings accounts for over 30% of total global final use and is responsible for 10% of global greenhouse gas emissions[1,2], which causes serious problems to both environment and economy. Statistics show that the annual building energy cost is over $430 billion in the U.S.[1,2]. Among this huge energy consumption, approximately 48% is for space heating and cooling. Moreover, because of climate change and population growth, it is predicted that heating and cooling energy consumption for buildings will grow by 79% and 83%, respectively, over the period of 2010–2050[3]. Therefore, achieving high energy efficiency in buildings with minimum carbon footprint has become an essential goal for sustainability and calls for innovation of science and technology[4]. One grand challenge is that most buildings are located in highly dynamic weather that can compromise the efficacy of passive solar heating or radiative cooling. These variations are both spatial and temporal, which include diurnal and seasonal fluctuation, climate zone dependence, energy price fluctuation, and so on[1–3]. For example, heating degree days and cooling degree days can commonly and quantitively describe the heating and cooling demands of buildings[5]. Figure 1a shows the annual heating and cooling degree days of 16 U.S. cities that represent the 16 climate zones. It can be found most cities need both heating and cooling throughout the whole year. Taking Durham, North Carolina as an example, the cooling consumption predominates from May to October, and the rest 6 months are heating-dominant (Fig. 1b). These statistics clearly manifest the need for smart and renewable indoor thermal environmental management that can switch between cooling and heating to cope with various situations and to achieve higher energy saving all year round.

To accomplish this goal, we resort to two infinite radiative heat source and cold source: The Sun (5800 K) and the outer space (3 K), respectively, to supply heating and cooling to buildings without using fossil fuels. For ideal daytime radiative cooling materials, the materials should have a high reflectance in 200–2500 nm and high emissivity in 8–13 μm[6]. For ideal solar heating, it is expected the material has high absorption in 200–2500 nm and low emissivity in >2500 nm[7]. Prior research efforts for both solar heating[8–11] and radiative cooling[12–29] have yielded both high technological performance and deep scientific understanding, which spans from a variety of fields, including materials science, photonics and plasmonics, and heat transfer. However, they are mostly static or quasi-dynamic devices, which cannot completely solve the dynamic heating and cooling demand problem effectively, especially in the daytime[12–29]. For the few prior reports of dynamic solar and mid-IR heat management, none of them demonstrate the ideal properties for both modes—selective absorber for solar heating and highly solar reflective layer for daytime radiative cool cooling (Supplementary Table 1). In other words, the heating/cooling performance was sacrificed for the dynamic tunability. In this work, we demonstrate the dual-mode smart heat managing device that possesses the ideal dual-mode optical properties and can achieve up to 71.6 W/m² of cooling power density and up to 643.4 W/m² of heating power density (over 93% of solar energy can be utilized) from experimental tests by optimizing the optical, mechanical, and heat transfer properties at various scales, ranging from nanoscale surface morphology to device-level design. We also performed the rigorous calculation of building energy efficiency that encompasses most of the major cities in the U.S. to create the energy saving map for various climate zones. The map shows our dual-mode device outperforms the solar-heating-only and radiative-cooling-only devices, which can save 19.2% of building heating and cooling energy on average.

## Results

**Concept of the dual-mode device.** As shown in Fig. 1c, the dual-mode heat managing device consists of a pair of rotary actuators or rollers and a thin-film polymer composite that has solar heating and radiative cooling functions side-by-side. In the cooling part of the material, sunlight is mostly reflected, and the thermal radiation to outer space through the atmospheric long-wave infrared transmission window (8–13 μm) is maximized, thereby achieving passive sub-ambient cooling that can contribute either directly to the wall/roof via heat conduction or to the heat exchanger that removes the heat from the chiller of air conditioning system. In the heating part, most of the solar energy is absorbed, and the radiation loss is strongly suppressed by the selective absorber, which results in high heat flux to the building envelope or the heat exchanger. When mode-switching is needed because of the change in weather or in energy balance, the actuators will pull the heating/cooling materials to move along the track system and expose the desired part of the materials to work in the ideal mode, and the rest of the material is rolled up and collected (Supplementary Movie 1). We point out there are four key points to successfully realize the dual-mode design: (1) The material should have excellent solar heating and radiative cooling properties to obtain high heating and cooling performance that is on par with most state-of-the-art solar heating and radiative cooling materials alone. (2) The material needs to have low thermal resistance to fully utilize the generated heating/cooling power or temperature difference. (3) The material needs to be flexible and lightweight for the rolling actuation process and durable to maintain the performance after cycles of switching. (4) The thermal contact resistance needs to be minimized between the soft and flexible material and the heated/cooled objects, i.e., roofs or heat exchanger. Although it is relatively straightforward to reduce the thermal contact resistance between stationary objects, it is not a trivial task for movable objects such as the heating/cooling material. We will explain how to achieve these four design requirements in the following paragraphs.

**Electrostatically controlled thermal contact.** For dual-mode year-round building energy saving, as mentioned above, one important aspect is to reduce the thermal contact resistance between dual-mode material and the underlying object, otherwise, the heating/cooling energy would not be transported and fully utilized by the building. Although there are many effective methods to ensure good thermal contacts, such as welding and thermal interface materials[30,31], these strategies are not applicable in our system because the material needs to be frequently attached/removed for mode-switching. Note that it is also not ideal to increase the material weight and resort to larger gravitational force to reduce the contact resistance because that will increase the thermal resistance of the material itself and the required energy for actuation (Fig. 1c). As a result, if the dual-mode material thin film is simply let in contact with the object, the wrinkled texture leaves significant amounts of air gap in between, and the contact resistance is extremely large (Fig. 2a, left). Choosing a polyimide (PI) film as the substrate can partially resolve the issue due to its capability to carry static charges (Fig. 2a, middle). To further boost the performance, we apply high voltage to the electrode and use the Maxwell pressure to accomplish strong yet reversible thermal contact between the material and the object (buildings or heat exchanger) (Fig. 2a, right). To visually demonstrate this point, we placed a radiative cooling material on a constant temperature copper plate and uses the infrared camera to record the temperature change under different applied voltages (Fig. 2b). The corresponding optical images are in Supplementary Fig. 1. Quantitatively, we use a

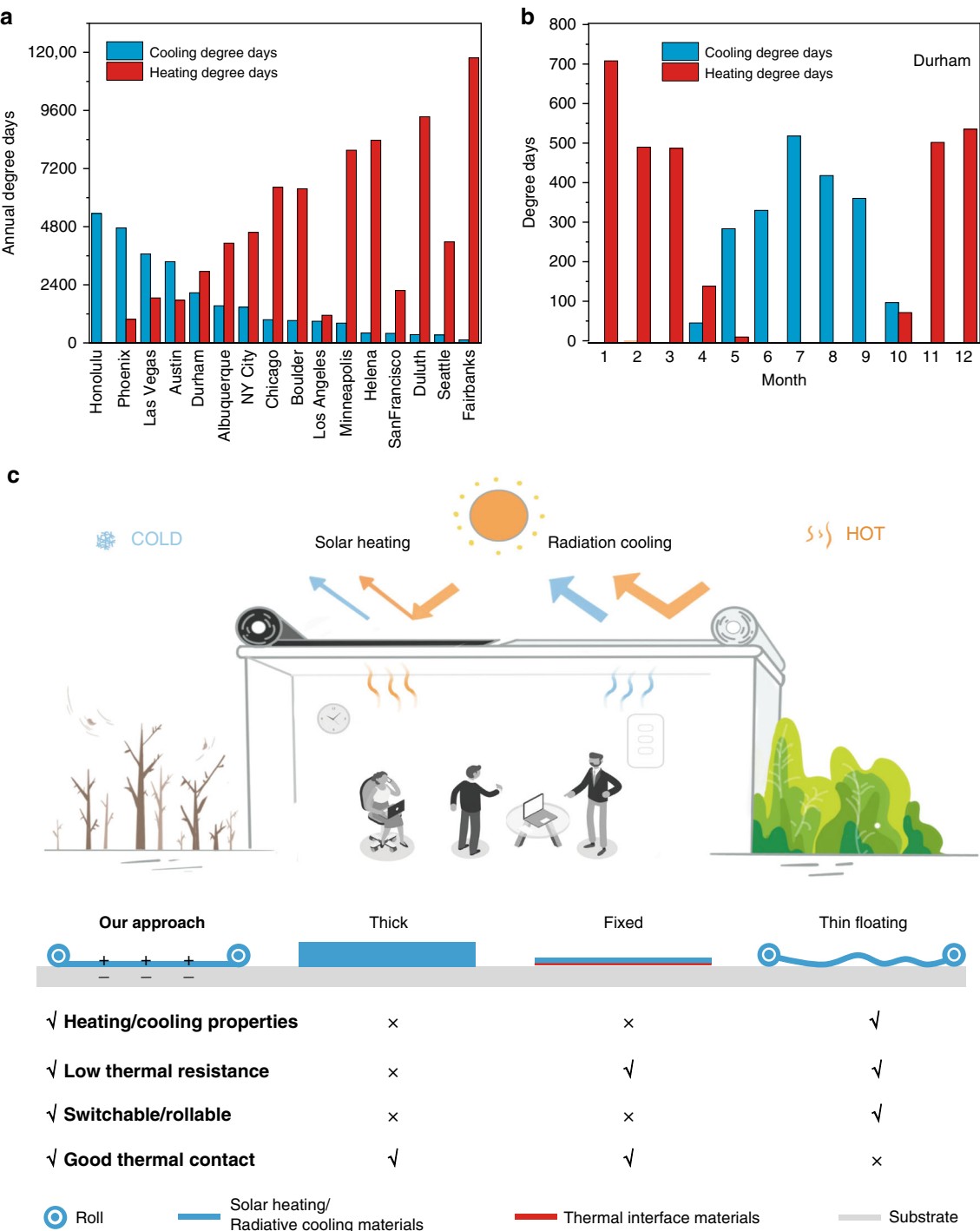

**Fig. 1 Concept of dual-mode device. a** Annual heating and cooling degree days of 16 U.S. cities that represent the 16 climate zones. **b** Heating and cooling degree days over 12 months in Durham, NC, USA. **c** Schematic of the dual-mode device at heating (left) and cooling (right) mode. The switchable building envelope can utilize both renewable heating and cooling sources. We identify the four criteria of the dual-mode device: heating/cooling optical properties, thermal resistance, rollability, and thermal contact.

carbon black reference on the plate to obtain the temperature difference between sample and substrate to calculate the contact resistance (more details can be found in "Methods"). As shown in Fig. 2c, the average thermal contact conductance significantly increased as the applied voltage increased. The Maxwell pressure increases not only the macroscopic contact area (Supplementary Fig. 1) but also the microscopic area[32], both of which result in the reduction of overall thermal contact resistance. The average thermal contact conductance can reach $9.5 \times 10^2$ W/m$^2$K at 2 kV

applied voltage., and the corresponding temperature difference is suppressed to about 0.4 °C (Supplementary Fig. 2). Note that even the voltage is high, the current is only about 0.07 mA, which is a safe current for the human body[33]. Moreover, as shown in Fig. 2d, the average thermal contact resistance between PI film and substrate is almost unchanged after removing the voltage source for 3 days (Ambient condition: 20 °C, 40% humidity) due to the high electrical resistivity and hydrophobicity of PI. It can be expected that this static electricity may not last for several

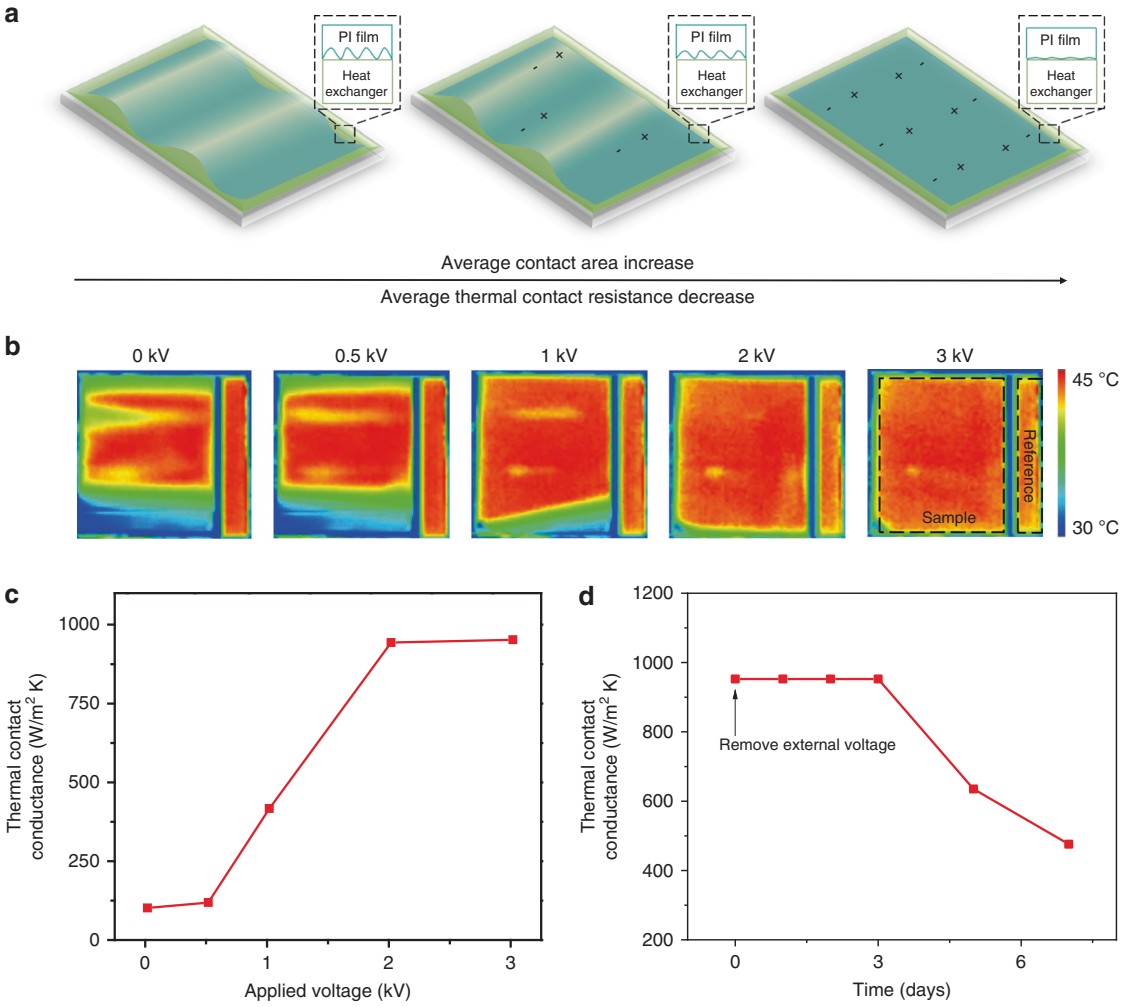

**Fig. 2 Reducing the average thermal contact resistance by the electrostatic effect. a** Schematic of dual-mode material morphology evolution as the function of surface static charges. As the static charge increases by either triboelectricity or applied voltage, the Maxwell pressure can increase both the macroscopic contact area and the local contact conductance, which significantly decreases the overall thermal contact resistance. **b** Thermography images of the cooling material on a constant temperature substrate after applying 0, 0.5, 1, 2, and 3 kV, respectively. **c** Average thermal conductance over applied voltage. **d** The average thermal conductance remains unchanged for 3 days even after the applied voltage is removed because of the strong tendency of the PI film to retain surface charges.

months, but in practical applications, the thermal contact can be rebuilt by periodic "charging", which is completed within a few seconds each time. This ability to maintain the static charge effect reduces the need to constantly supply high voltage and therefore enhances the device operation efficiency and lifetime (analysis of the impact of dirt and humidity can be found in Supplementary Note 1). When switching is needed, a small reverse bias can be applied to offset the static charges and release the dual-mode material (Supplementary Movie 1).

**Optical properties of heating and cooling materials**. For dual-mode year-round building energy saving, good thermal contact is necessary but not sufficient conditions. The other important aspect is to design a rollable film with low thermal resistance and high heating/radiative cooling performance. Figure 3a illustrates the structure of the dual-mode heating/cooling material. Besides the strong tendency to retain surface static charges, PI film was selected as the substrate also because of its excellent flexibility (for mode-switching and rolling), smooth surface (for a smooth metallic back reflector in cooling mode and for lowering the thermal contact resistance), good mechanical property (for reducing the thickness and the thermal resistance of substrate

itself). Figure 3b shows the drastically different visual appearance of the two parts of the material: the heating part is black for sunlight absorption, and the cooling part is mirror-like for sunlight reflection. In the heating part, zinc film with copper particle was deposited as heating material due to its excellent selective absorption property[7]. The morphology and composition of heating material were charactered by scanning electron microscopy (SEM) and X-ray photoelectron spectroscopy (XPS), as shown in Fig. 3c, d. It can be found that clusters of approximately 1 μm in size composed of copper and copper oxide nanoparticles were uniformly deposited on Zn film (the optimized experiment can be found in Supplementary Note 2). Here, copper is used as the electrode for zinc electrodeposition to produce a uniform film (Supplementary Fig. 3), and it also serves as the electrode to apply electrostatic charges. On top of the PI film, a silver film of 300 nm thick is deposited, followed by polydimethylsiloxane (PDMS). This part of the material is designed for cooling because silver can reflect the majority of solar radiation and PDMS has excellent transparency in the visible regime and high thermal emissivity in the mid-infrared regime. As shown in Fig. 3e, f, it can be found that, with the increase of thickness of PDMS, the transmission of the visible regime is almost unchanged, and the absorption of

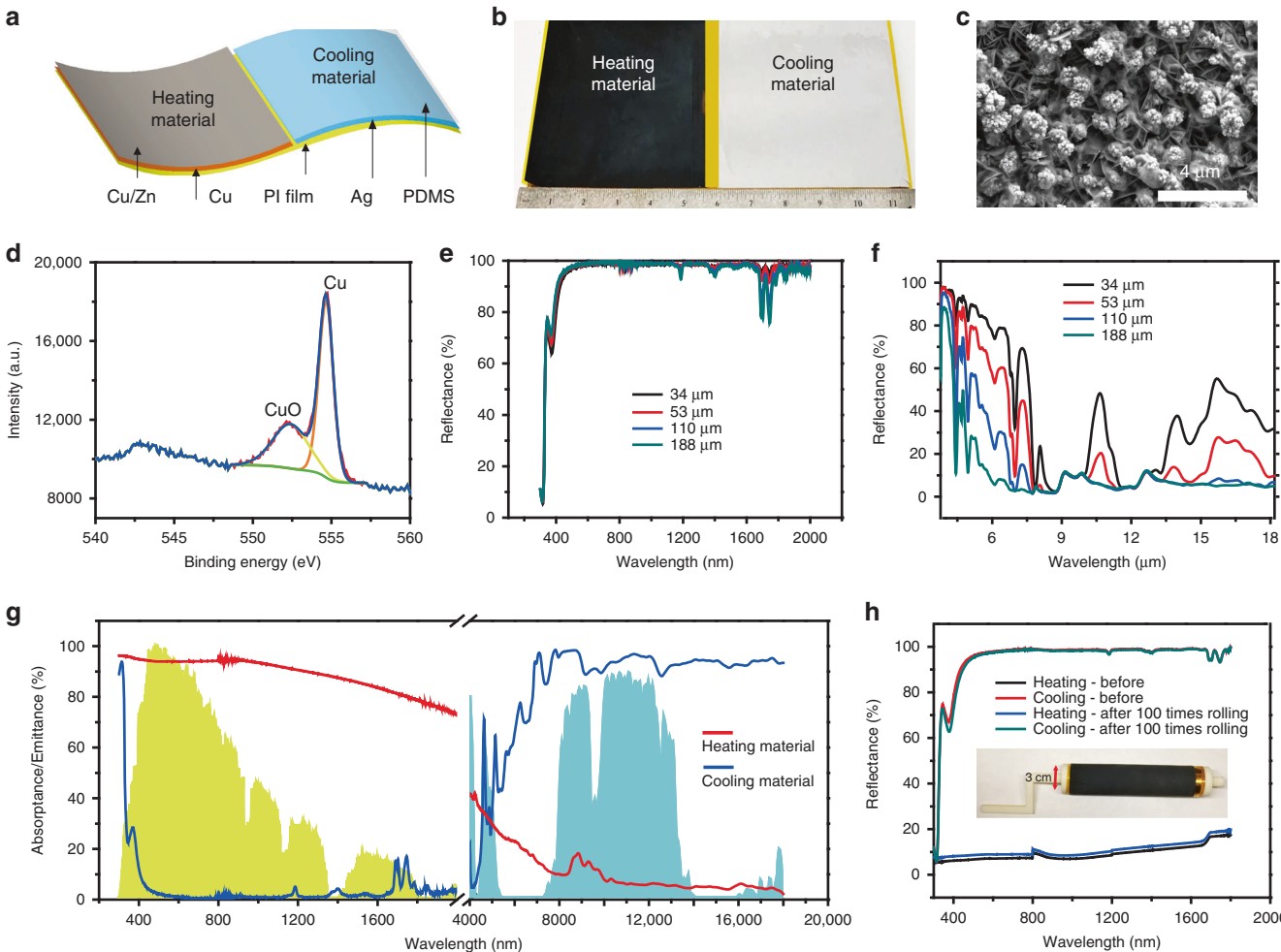

**Fig. 3 Dual-mode material structure and properties. a** Structure of dual-mode heating/cooling material. Polyimide (PI) is the common substrate. Above the PI substrate, different configurations are used for the two modes. For cooling mode, Ag film is the electrode for exerting the electrostatic force (Maxwell pressure) and for reflecting the solar radiation. The top PDMS layer is visibly-transparent and infrared-emissive for radiative cooling. For heating mode, Cu is the electrode for supplying static charge, and the Cu/Zn is the plasmonic selective absorber. The thickness of PDMS, Cu/Zn, Ag, Cu, and PI film are 110 μm, 1 μm, 300 nm, 300 nm, and 25 μm, respectively. **b** The photo of the dual-mode material shows the different visible appearance of the heating/cooling parts (**c**), SEM image of the heating material. **d** XPS spectrum of a copper particle on heating material. **e**, **f** Visible, near-IR, and mid-IR reflectance spectra of cooling materials of different thicknesses. **g** Absorptance/emittance of dual-mode material. Solar spectrum (yellow shaded area), and atmospheric transmittance window (green shaded area) are plotted for references. **h** The reflectance of heating and cooling material before/after 100 times rolling testing. The inset is the photo of the sample under testing.

mid-infrared increases until 110 μm. Hence, 110-μm-thick PDMS was selected for further testing. After parameter optimization (Fig. 3e, f), as shown in Fig. 3g, the cooling part has 97.3% reflectance in the wavelength range of 300–2000 nm and 94.1% emissivity in 8–13 μm. For the heating part, 93.4% absorption from 300–2000 nm and 14.2% emissivity were achieved. Moreover, the sample shows high mechanical flexibility and robustness. As shown in Fig. 3h, the performance of our sample is almost unchanged after 100 times rolling test. These standalone heating/cooling properties lay the foundation for our high-performance dual-mode heat management building envelope device.

**Outdoor performances of the dual-mode device.** To measure both the solar heating and radiative cooling heat flux of the dual-mode material, the Peltier-based performance measurement system was set up. The dual-mode testing system consists of mainly four parts: Peltier temperature control feedback system, data acquisition system, high voltage power supply, and ambient condition measurement system (Fig. 4a). As shown in Fig. 4b, the temperature control system uses the Peltier device to supply

heating/cooling power to the copper plate, and a PID control program is employed to minimize the temperature difference between the copper plate and the ambient temperature, both of which are measured by thermistors connected to the data acquisition system and laptop. This method is designed to minimize the convective heat loss to/from the ambience[14,20]. At a steady state, the thermoelectrically-supplied heat flux equals the solar heating (downward heat flux) or the radiative cooling (upward heat flux), which is measured by the heat flux sensor between the Peltier device and the copper plate. As shown in Fig. 4c, d, the switching process between heating and cooling can be achieved by motors (more details can be found in Supplementary Movie 1) or manually (see Fig. 4a). The outdoor experiment was performed on the campus of Duke University at Durham, North Carolina, on October 24, 2019. The solar power intensity, humidity, and ambient temperature are measured in real-time to calculate the model values to predict the heating/cooling heat fluxes (Supplementary Fig. 4). The numerical model is an essential tool to calibrate the performance with respect to the local weather condition.

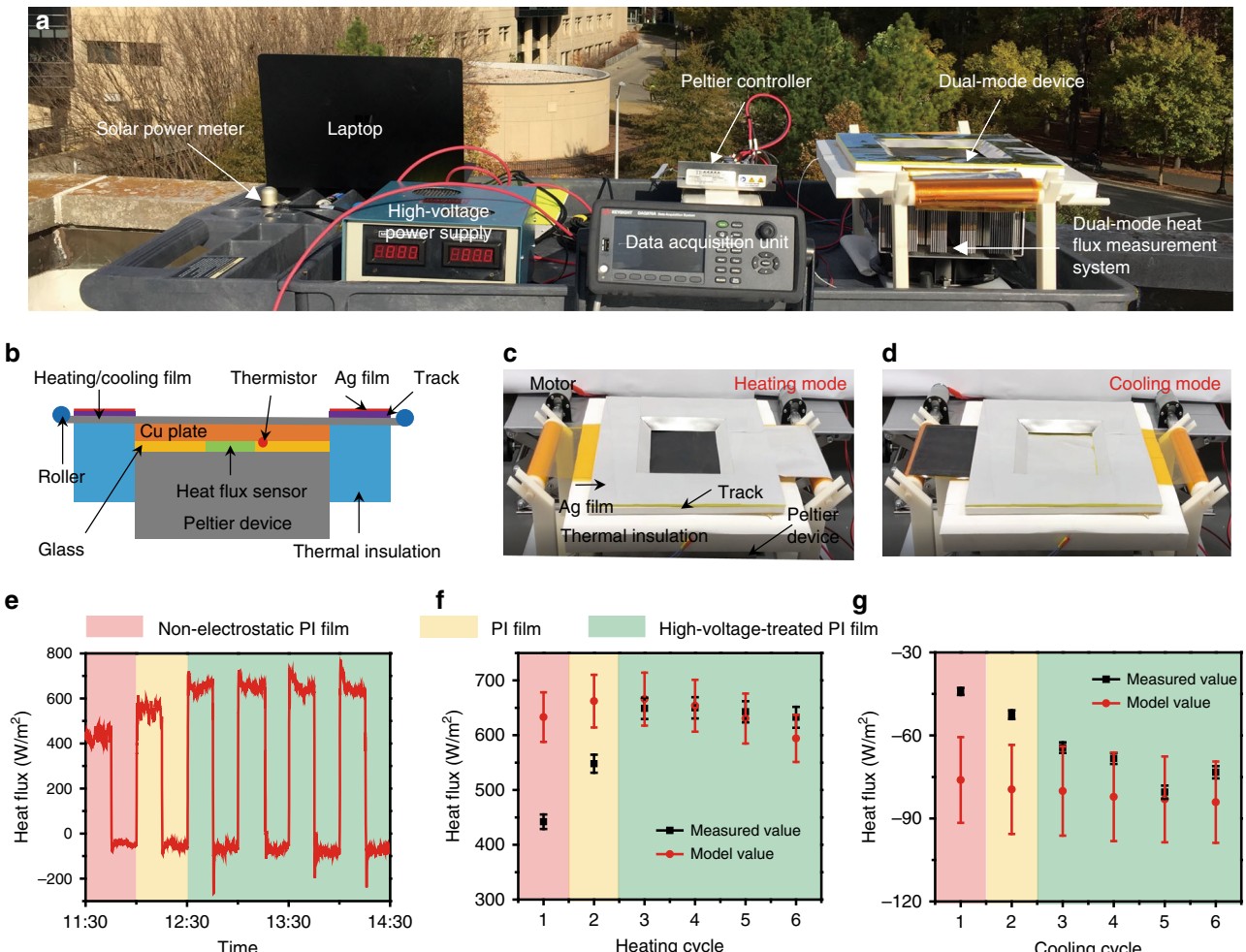

**Fig. 4 Testing system and outdoor real-time performance of the dual-mode device. a** Photo of the overall testing system. **b** Schematic and **c**, **d** photo of the dual-mode heat flux measurement device with heating mode and cooling mode. **e** Measured heat flux during the testing period. Positive values represent heating. For comparison, the first cycle (red area) was performed using a non-electrostatic PI film, the second cycle (yellow area) used PI film without external voltage, and the rest of the cycles (green area) used PI film with external voltage supply. The efficacy of Maxwell pressure-induced thermal contact is clearly shown. **f** Average solar heating heat flux over cycles. **g** Average radiative cooling heat flux over cycles. The error bars reflect the measurement accuracy.

The heating and cooling performance of the dual-mode device is tested for six heating and cooling cycles (15 min per cycle), as shown in Fig. 4e. The positive heat flux represents heating, and the negative flux represents cooling. To demonstrate the importance of thermal contact conductance in the practical heating/cooling conditions, we fabricated three samples for testing: the gold-coated PI film without electrostatic charges (shown in red area), normal PI with zero voltage (yellow area), and normal PI film with 2 kV high voltage treatment (green area). As the electrostatic force increases, both heating and cooling heat fluxes increase, manifesting the impact of thermal contact conductance on the performance. The switching happens almost instantaneously, and the system can return back to thermal equilibrium within less than 100 s. Because of the small thickness and low heat capacity of the dual-mode material, there is very little thermal inertia to overcome, and the transient heat conduction can quickly propagate to the object (in this case, the heat flux sensor). In heating cycles, as shown in Fig. 4f, the average heat flux of non-electrostatic PI and zero-external-voltage PI only achieve $442 \pm 13.3$ and $548 \pm 16.4$ W/m$^2$, which is 70% and 83% of the model values ($633 \pm 45.2$ and $662.2 \pm 48$ W/m$^2$, respectively). After applying high voltage, the device generates an average heating power of 643.4 W/m$^2$), which means over 93% of

solar energy is utilized. Similarly, in cooling cycles, as shown in Fig. 4g, the average cooling heat fluxes of non-electrostatic PI and zero-external-voltage PI are $44.1 \pm 1.3$ and $52.5 \pm 1.6$ W/m$^2$, which are 58% and 66% of the model values ($76.1 \pm 15.5$ and $79.5 \pm 16.1$ W/m$^2$, respectively). After applying high voltage, the measurement (average cooling power is about 71.6 W/m$^2$) and model match well. More details about model calculation can be found in Supplementary Note 3. This experiment clearly demonstrates the effectiveness of using electrostatic force to build the thermal contact between the dual-mode material and the object and how it can boost the performance approaching the theoretical prediction.

**Building energy simulation.** To quantitively predict the potential impact of our dual-mode device on building energy efficiency, we used EnergyPlus together with empirical material property data to calculate the year-round energy saving for heating-only, cooling-only, and dual-mode building envelopes. 16 cities were selected to represent the 16 climate zones in the U.S[2]: Albuquerque, Austin, Boulder, Chicago, Duluth, Durham, Fairbanks, Helena, Honolulu, Las Vegas, Los Angeles, Minneapolis, New York City, Phoenix, San Francisco, and Seattle. The corresponding energy-saving

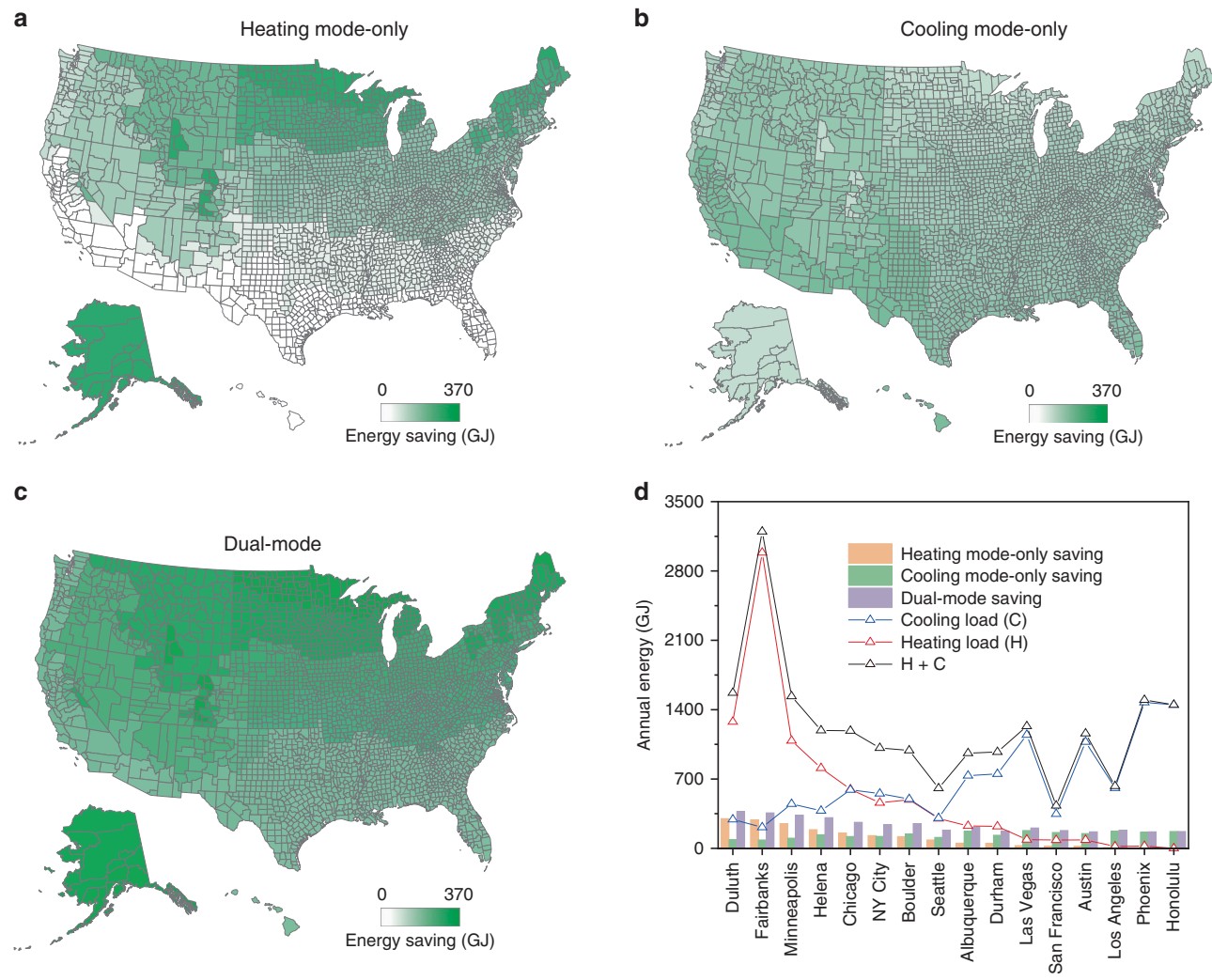

**Fig. 5 Modeled annual energy saving of different energy-saving modes.** Building energy-saving map for all U.S. climate zones with **a** only heating mode, **b** only cooling mode, and **c** dual-mode, **d** annual averaged energy saving at different modes and energy loads over different cities.

maps of heating-only and cooling-only are shown in Fig. 5a, b, respectively (More details about calculation can be found in Supplementary Note 3). These U.S. building energies saving maps convey several aspects of information. Firstly, solar heating is more beneficial in the north, and radiative cooling benefits more in the south. This is understandable based on the latitude dependence of climate. Secondly, radiative cooling saving is slightly larger than solar heating saving. This can be attributed to: (1) When the sunlight is relatively abundant, the required heating load is small (Noting that we only considered real-time usage, and did not consider any energy storage technology. It is expected that if some storage technologies can be combined, the energy savings will be significantly improved[22].). (2) The radiative cooling technology can provide cooling power throughout the entire 24 h when space cooling is needed. The energy-saving map of the dual-mode building envelope is then calculated and plotted in Fig. 5c. It can be seen that the dual-mode device has significant advantages in almost all climate zones in the U.S. compared with single-mode devices. To further manifest the overall impact to the U.S. building energy efficiency, we calculated the annual averaged energy consumption saving in GJ (Fig. 5d, the more details about the calculation process can be found in Supplementary Note 3). The baseline of annual energy consumption for building heating and cooling is 548 and 681 GJ, respectively. The calculation indicates that the dual-mode device can save 236 GJ (19.2% of the

heating and cooling energy), which is 1.7 times higher than cooling-only (138 GJ) and 2.2 times higher than heating-only (106 GJ) devices.

## Discussion

In this work, we experimentally and computationally demonstrated the heating/cooling performance of the smart dual-mode building envelopes, which is the outcome of the rational design of optical, mechanical, and heat transfer properties. Such dynamic tunability will gain more and more significance as the renewable yet intermittent energy resources, such as solar and wind power, are being incorporated into the electric grid. On the other hand, climate change can also aggravate the spatial and temporal climate fluctuation in both frequency and magnitude, which calls for more adaptive building energy efficiency solutions. Together with our heating/cooling dual-modes device, we envision devices with more energy modes, source-tracking functions, system-level optimization algorithm, or smart grid integration, will form the new design paradigm of zero-energy buildings.

## Methods

**Fabrication of heating material and cooling.** Polyimide (PI) film (width (W): 13 cm, length (L): 29 cm, thickness (T): 25 μm) was selected as the substrate. Silver film (W: 12.5 cm, L: 14 cm, T: 300 nm) and copper film (W: 12.5 cm, L: 14 cm, T: 300 nm) were deposited onto the PI film side-by-side using the evaporator (Kurt

Lesker PVD 75). The silver film was coated with a layer of PDMS of about 110 µm thick as the cooling material. On the copper film, a layer of zinc film of 1 µm thick was electrodeposited (voltage: 2 V, anode: zinc metal, electrolyte: 0.25 M $ZnSO_4$ (aq)), followed by galvanic replacement reaction with 0.12 mM $CuSO_4$(aq), and the heating material was obtained after deionized water washing and drying.

**Characterizations**. The morphology of the heating material was characterized by SEM (FEI Apreo) with the beam voltage/current of 5 kV/25 pA. The chemical composition of the heating material was characterized using XPS (Kratos Analytical Axis Ultra), equipped with a monochromatic Al Ka X-ray source. The reflectance of cooling material and heating material was measured by the UV–visible–near-infrared spectroscopy spectrometer with a calibrated $BaSO_4$ integrating sphere (300–2000 nm, Agilent technologies, Cary 6000i) and the Fourier Transform Infrared spectrometer with a diffuse gold integrating sphere (4–18 µm, Thermo Scientific, iS50). The thermal images in our experiment were captured by the FLIR E60 infrared camera.

**Commercial building heating and cooling model**. *EnergyPlus* version 9.2 was utilized to predict energy consumption and saving with different boundaries. Sixteen cities were selected to represent the 16 climate zones in the U.S.: Albuquerque, Austin, Boulder, Chicago, Duluth, Durham, Fairbanks, Helena, Honolulu, Las Vegas, Los Angeles, Minneapolis, New York City, Phoenix, San Francisco, and Seattle. The post-1980 medium office model defined by the U.S. Department of Energy[34] was adopted for the calculation. The model building has three stories, and the roof area is 1660 m². For temperature boundary conditions, monthly internal temperature has been set by the medium commercial building model. Hourly weather data in the typical meteorological year[35] was used as the external environment boundary condition. Baseline energy consumption for heating and cooling was established by running this model across the U.S. Cooling load power per month ($P_{load}$) could also be calculated in this process.

**Thermal contact conductance measurement**. Supplementary Fig. 5 shows the equipment to quantitatively characterize the thermal contact conductance. From the top surface, it involves reference (carbon glue of 80 µm thick)/sample, copper plate (Width: 5 cm, Length: 5 cm, Thickness: 1.5 mm), heat flux sensor of 1 mm thick (Electro Optical Components, Inc., A-04457) surrounded by glass with the same thickness, PID-controlled Peltier device (TE Technology Inc., TC-36-25). Thermal grease (Dow Corning, 340) was applied to ensure good thermal contact among Cu plate, glass slides, heat flux sensor, and Peltier device. The sidewall of the equipment is wrapped with polyurethane foam of about 5 cm thick to avoid heat loss. A calibrated thermal camera (FLIR E60) was used to record the steady-state temperature of reference and sample and therefore obtain temperature difference, $\Delta T$. Meanwhile, the heat flux sensor was used to measure the heat flux ($q$) in W/m². Assuming the temperature of reference is the same as the copper plate and neglect the thermal resistance of the sample itself, we can calculate the thermal contact conductance ($h_c$) based on $h_c = q/\Delta T$.

**Measurement of radiative cooling and solar heating power**. The apparatus for measuring solar heating and radiative cooling heat flux is similar to the one used to measure thermal contact conductance but with a few operational differences. The temperature of the copper plate is kept the same as the ambient temperature by using the PID program, so the convective heat loss to/from the ambiance can be minimized. Both temperatures are measured by the thermistors (TE Technology Inc., TC-36-25). At steady-state, the supplied heating or cool power compensates for the radiative cooling or solar heating, and the heat flux sensor measures the corresponding cooling (upward heat flux) or heating (downward heat flux) power in W/m².

Calibration of the testing apparatus was performed every time before measurement. As shown in Supplementary Fig. 6a, a heater was put on the top surface of the copper plate which is covered by a layer of polyurethane foam of 3 cm thick to avoid heat loss. The other part is the same with the testing system. Supplementary Fig. 6b shows the voltage reading of the heat flux sensor when we supply different heating power. The system quickly reaches a steady state after <100 s. As shown in Supplementary Fig. 6c, due to the difference in thermal resistance between the heat flux sensor and the glass slide, the nominal heat flux is 1.40 times larger than that measured by the heat flux sensor, which is independent of the applied power. In other words, the heat transfer pathway from the top copper plate through the heat flux sensor to the bottom Peltier device is more resistive than that pathway through glass slides per unit area. This ratio is steady-state and is independent of applied heating power, and it is used as the rescaling factor for outdoor measurement for both radiative cooling and solar heating.

## Data availability
The original data that support the findings of this study are available from the corresponding author upon request.

## Code availability
The code for the device performance and building energy consumption model can be made available upon request.

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

## Acknowledgements

We thank Pratt School of Engineering at Duke University for the funding support and Dr. Birgit Andersen (North Carolina State University) for the help of UV–vis measurement. We also thanks Dr. Jyotirmoy Mandal (University of California Los Angeles) and Dr. Dongliang Zhao (Southeast University) for discussion.

## Author contributions

P.C.H. and X.Q.L. conceived and planned the study; X.Q.L. performed the experiments; G.T., P.C.H., X.Q.L., B.W.S., C.X.S., A.N., and H.M.F. performed the simulation. X.Q.L., P.C.H., G.T., B.W.S., C.X.S., and Y.C.P. performed data analysis; X.Q.L., P.C.H., B.W.S., and C.X.S. wrote the paper. All authors discussed the results and approved the final version of the paper.

## Competing interests

P.-C.H. and X.Q.L. have a U.S. patent application (No. 62/942,257) related to this work. The remaining authors declare no competing interests.
