## [Peer Review File · Nature Communications]

Reviewers' comments:

Reviewer #1 (Remarks to the Author):

Radiative cooling is an emerging technology for energy sustainability. This work presents a dual-mode device for energy saving in HVAC systems. An electrostatically-controlled thermal contact film is introduced for better thermal conductance. The solar heating/radiative cooling mode could be adaptively switched to different environment to save energy. The general idea is interesting and well presented. However, I have concerns at the following points:

1. The absorption spectrum of the heating material in the mid-infrared regime is missing. By suppressing the infrared emission, is it possible for the heating mode to obtain a higher temperature for more efficient heating?
2. The outdoor test is confusing. From the schematic of the setup shown in Fig. 4b, it seems that the system did not even implement the pre-mentioned dual-mode device into the test. Instead, only a copper plate was used in those tests. How did the authors realize the mode-switching using the copper plate? It is necessary to identify the functionality (both heating mode and cooling mode) of the proposed device in the field tests.
3. Figs. 4f & 4g: The heat flux in different cycles are altering from positive to negative as the mode switching. As presented in individual cycles in figs. 4f and 4g, the presented heat flux is a discrete value rather than a continuously measured data set. However, the colored background and connected line gives readers an opposite impression that the heat flux follows the same time track in fig 4e, which is misleading.
4. In the last line of page 16, the authors claimed 51.4 GJ can be saved by the proposed dual-mode device. It is unclear how did this number was calculated. Details are required to justify whether this estimation is convincing.
5. The required voltage to induce the high-thermal contact is in kV level, which seems too high for civil applications. Besides, the duration after charge removal in fig 2d is too short for an HVAC application that usually last for months. Therefore, I have major concern on the practicability at this point.
6. The discussion in fig. 3c is insufficient. Although the authors try to explain the figure by referring fig. S7, it is still unclear regarding how the morphology or grain size affect the optical feature of the heating material.

Reviewer #2 (Remarks to the Author):

Review of “Integration of radiative cooling and solar heating for year-round energy saving of smart building envelopes” (NCOMMS-20-08399) by Xiuqiang Li, Bowen Sun, Chenxi Sui, Nandi Ankita, Haoming Fang, Yucan Peng and Po-Chun Hsu.

The authors propose a dual-mode device and demonstrate that it is capable of switching between radiative cooling and solar heating for year-long energy savings both in the heating and cooling conditions. Designed actuators will physically rotate the cooling/heating films along a track system to switch between two different films. The work focuses on reducing the interfacial thermal resistance between the functioning layer and the substrate surface by applying a high voltage of 2 kV. The overall energy savings are estimated and compared with heating-only or cooling-only devices.

A dual-mode device is beneficial to switch between the radiative cooling mode or solar heating mode at different seasons. However, we believe that the work is an incremental advance and does not have the novelty that is suitable for publication in Nature Communications. In addition, there are several concerns that need to be addressed.

1. The device relies on 2 kV to pull down the cooling or heating layers on the substrate to reduce interfacial thermal resistance. Although the device does not require continuous voltage application, the authors should elaborate on an economical solution to safely generate the required 2 kV high voltage within the device. It is critical to include the energy/equipment cost to initiate the 2 kV voltage that pulls down the cooling/ heating film. Additionally, the authors should include the installation and maintenance cost as well.

2. The work does not address some of the reliability issues when dirt or water gets into the interface between the PI film and substrate, or inside the track system. If there is dirt accumulating on the substrate beneath the films under windy conditions, the high voltage may not be sufficient to reduce the interfacial thermal resistance. Furthermore, water or humidity is also known to reduce the effect of static electricity. These are especially important when the films are exposed to the ambient air.

Reviewer #3 (Remarks to the Author):

This paper presented the technical apparatus and performance of a novel dual-mode radiant heating and cooling panel which can, most critically, reject heat to the ambient sky even during sunny day-time conditions. The authors provide a validation of experimental data, followed by calibration of an empirical model, followed by the use of this model (coupled with a building energy model) to assess the potential cooling and heating savings offered by the technology across the United States.

My comments regarding the paper are provided in the following sections:

About the Physical apparatus

- I do not pose sufficient expertise in materials engineering to understand and critique the methods, results and discussion surrounding the physical performance of individual panels. I trust this will be captured by other reviewers. However, I do not perceive the technical description and analysis of the panel itself to be the core proposed novelty of the paper, nor the means by which the authors demonstrate the proposed impact of their innovation.

About the climate analysis undertaken

- I feel the main overarching disadvantage of the manuscript is an apparent oversimplification, or misrepresentation, of data and results that aim to assess the viability of the proposed technology for heating and cooling across the US. This begins early in the manuscript in the presentation of a classification of cities with regards to 'heat management monotony'. I'm not familiar with such a term, but the analysis summarised in Figure 1a is counter-intuitive and seems to disagree with supplemental data provided by the authors in Figure S4. For example, from Figure 1a, Seattle appears to be a more cooling dominant city than San Francisco, but clearly from Figure S4, San Francisco very clearly possesses a larger and more persistent cooling load than Seattle, and Seattle conversely possesses a much larger heating load. The authors should consider reverting to more widely-understood metrics for heating/cooling classification of cities such as Heating Degree Days and Cooling Degree Days.

About the predictive model of radiant cooling

- The model assumes that the main intermediary between the panel and outer space is the atmosphere. This is true, but it would appear that the chosen model assumes a clear sky exists at all times and so transmissivity is only affected by the vertical water column. As it is the intent of this paper to examine the applicability of the radiant cooling panel across the continental US on an annual basis, this assumption cannot hold. Accounting for the effect of cloud cover is challenging, but it is necessary to do so. Not only do clouds change the radiant interaction between the panel and the edge of the atmosphere, depending on their altitude and temperature, they may be a source of radiation in their own right.

Spatial obstructions and view factors

- The model assumes that the effective roof area that can be covered by the proposed radiant panel system may be 60%. No further information is given regarding the effective sky view factor of rooftop systems. This surely cannot be 1 due to the diversity of likely obstructions that typical rooftops face, from surrounding vegetation to surrounding buildings.

Other viable interactions between the rooftop and the sun.

- During summer daytime periods, it would appear that there is still an argument to prefer utilisation of roof area for solar PV generation. Electricity generation can serve multiple uses, from building to (increasingly) transport energy services. Modern solar PV systems may generate well over 150 W/m² in

electricity. Under the same form of system modelling as applied in this work, one could envisage this solar PV electricity serving a heat pump-cooling system with a COP in excess of 3. From an energy consumption mitigation standpoint, this would seem to be a far more effective solution for reducing current electricity usage for space cooling across the United States. On the other hand, an understated advantage of the proposed solution is that it can operate under its 'cooling' mode during both daytime and nighttime. This does pose questions, however. Would the system operate at times of night where an immediate indoor cooling demand does not exist, and therefore it would be valuable to store chilled water? If so, how would such a system perform, particularly with respect to exergy? In general, the underdeveloped nature of the energy model, from unclear assumptions made regarding building typologies (is an 1660 m² office building meant to be representative of the US building stock?), to unclear interactions between the radiant cooling panel and dynamic weather as discussed, would suggest more thoughtful analysis or explanation of this part of the provided work is needed.

Ultimately, I would propose that, if building energy simulation is to be relied on to forecast the viability of the proposed radiant heating / cooling system to entire building stocks, greater consultation of prior research from the building simulation is needed, as is adoption of existing best practises with regards to thermal energy systems modelling.

Response to reviewer #1

Radiative cooling is an emerging technology for energy sustainability. This work presents a dual-mode device for energy saving in HAVC systems. An electrostatically-controlled thermal contact film is introduced for better thermal conductance. The solar heating/radiative cooling mode could be adaptively switched to different environment to save energy. The general idea is interesting and well presented. However, I have concerns at the following points:

Our reply:

We thank the reviewer very much for his/her positive report and constructive suggestions. In the following sections, we would address the questions point by point.

1. The absorption spectrum of the heating material in the mid-infrared regime is missing. By suppressing the infrared emission, is it possible for the heating mode to obtain a higher temperature for more efficient heating?

Our reply:

Thank you for pointing out this omission. We have added it to the revised Fig. 3g (see Fig. R1 below as well).

Fig. R1. Dual-mode material structure and properties. **a**, Structure of dual-mode heating/cooling material. Polyimide (PI) is the common substrate. Above the PI substrate, different configurations are used for the two modes. For cooling mode, Ag film is the electrode for exerting the electrostatic force (Maxwell pressure) and for reflecting the solar radiation. The top PDMS layer is visibly-transparent and infrared-emissive for radiative cooling. For heating mode, Cu is the electrode for supplying static charge, and the Cu/Zn is the plasmonic selective absorber. The thickness of PDMS, Cu/Zn, Ag, Cu, and PI film are 110 μm , 1 μm , 300 nm, 300 nm, and 25 μm , respectively. **b**, Photo of the dual-mode material shows the different visible appearance of the heating/cooling parts **c**, SEM image of the heating material. **d**, XPS spectrum of copper particle on heating material. **e** and **f**, Visible, near-IR, and mid-IR reflectance spectra of cooling materials of different thicknesses. **g**, Absorbance/emittance of dual-mode material. Solar spectrum (yellow shaded area), and atmospheric transmittance window (green shaded area) are plotted for references. **h**, Reflectance of heating and cooling material before/after 100 times rolling testing. The inset is the photo of the sample under testing.

2. The outdoor test is confusing. From the schematic of the setup shown in Fig. 4b, it seems that the system did not even implement the pre-mentioned dual-mode device into the test. Instead, only

a copper plate was used in those tests. How did the authors realize the mode-switching using the copper plate? It is necessary to identify the functionality (both heating mode and cooling mode) of the proposed device in the field tests.

Our reply:

Thank you very much for your constructive suggestion. In fact, as shown in Fig.4 c and d, the switching process between heating and cooling can be achieved by motors or manually (see Fig. 4a). To make these points clear, we have corrected the schematic (Fig. 4b) and added the device photos in revised Fig. 4 c and d (see Fig. R2 below as well).

Fig. R2. Testing system and outdoor real-time performance of the dual-mode device. a, Photo of the overall testing system. **b,** Schematic and **c, d** photo of the dual-mode heat flux measurement device with heating model and cooling model. **e,** Measured heat flux during the

testing period. Positive values represent heating. For comparison, the first cycle (red area) was performed using non-electrostatic PI film, the second cycle (yellow area) used PI film without external voltage, and the rest of the cycles (green area) used PI film with external voltage supply. The efficacy of Maxwell pressure-induced thermal contact is clearly shown. **f**, Average solar heating heat flux over cycles. **g**, Average radiative cooling heat flux over cycles. The error bars reflect the measurement accuracy.

3. Figs. 4f & 4g: The heat flux in different cycles are altering from positive to negative as the mode switching. As presented in individual cycles in figs. 4f and 4g, the presented heat flux is a discrete value rather than a continuously measured data set. However, the colored background and connected line gives readers an opposite impression that the heat flux follows the same time track in fig 4e, which is misleading.

Our reply:

Thank you very much for your constructive suggestion. We have removed the continuous line in Figs. 4f and 4g (see Fig. R2 f and g).

4. In the last line of page 16, the authors claimed 51.4 GJ can be saved by the proposed dual-mode device. It is unclear how did this number was calculated. Details are required to justify whether this estimation is convincing.

Our reply:

We appreciate the comments. Because Reviewer#3 also commented on the building energy saving calculation and suggested refining the calculation by taking more practical complexities into account, we combined both reviewers' comments and redid the calculation with a more detailed method explanation. We collaborate with Prof. Gang Tan (Department of Civil & Architectural Engineering, University of Wyoming, USA) for his expertise in building energy engineering can help us improve the scientific rigorousness in the energy saving calculation. The details are listed below and also in Supplementary Information Note 3 (Page 11-16).

System cooling-mode energy saving:

Potentially, there are quite many application methods for the proposed dual-mode radiation heating and radiative cooling materials in buildings. We demonstrate an example application of integrating the material with building envelopes to provide space heating and cooling energy using heat exchangers. At this system level application, a comprehensive integrated analysis of the proposed device and the subject building is needed, which creates hourly performance simulation for the 16 cities throughout a whole year. In addition, in order to evaluate the energy savings of the dual-mode device under real application condition, the typical meteorological year (TMY3) (National Solar Radiation Data Base, 1991-2005 Update: Typical Meteorological Year 3; https://rredc.nrel.gov/solar/old_data/nsrdb/1991-2005/tmy3/. [Accessed May 28, 2020]) weather data are used, and the impacts of the humidity and the clouds on cooling capability are evaluated. Therefore, the following calculation algorithms (eq.1 – eq.6) are selected to estimate the cooling power with effective atmospheric emissivity (ϵ_{atm}):

$$P_{cooling\ power} = P'_{rad} - P'_{atm} \quad (1)$$

$$P'_{rad} = A\epsilon_{film}\sigma T_{film}^4 \quad (2)$$

$$P'_{atm} = A\epsilon_{film}\epsilon_{atm}\sigma T_{amb}^4 \quad (3)$$

where ϵ_{film} and T_{film} are the emissivity and surface temperature of the film, and A is the area. The ϵ_{atm} is given by⁵,

$$\epsilon_{atm} = \epsilon_{atm,c} (1 - 0.78CF) + 0.38CF^{0.95} RH^{0.17} \quad (4)$$

$$\epsilon_{atm,c} = 0.618 + 0.056 \sqrt{P_w} \quad (5)$$

$$P_w = P_0 \exp [(cT_d) / (T_d + T_0)] \quad (6)$$

Where $\epsilon_{atm,c}$ is the effective sky emissivity under clear skies, CF is the cloud fraction, RH is the ambient relative humidity, and P_w is the ambient water vapor partial pressure, T_d is the dew point,

$P_0 = 610.94$ Pa, $c_T = 17.625$, and $T_0 = 243.04$ °C. The hourly values of these weather-related parameters can be obtained from TMY3 weather data.

With heat transfer medium such as water flowing in the heat/cold exchangers or collectors, the cold water will possess varied temperatures with environmental weather changes. Lower temperature water can be directly used for space cooling through indoor systems such as radiant cooling ceilings, which commonly adopts a fluid temperature of 13-18 °C (Fernandez, N., Wang, W., Alvine, K., Katipamula, S. Energy savings potential of radiative cooling technologies. PNNL-24904, (2015).) in order to avoid surface condensation. In contrast, when the water temperature is higher than this range, the radiative cooling cold water can be supplied to air conditioner side and cool the condenser side to achieve higher efficiency (details seen in the next section). In other words, the cooling power is to directly cool the building spaces when the temperature is below 18 °C, and to cool the air conditioner condenser when the temperature is above 18 °C. Therefore, when the temperature is below 18 °C, that particular hour's $P_{cooling\ saving}$ is calculated by,

$$P_{cooling\ saving} = MIN (P_{cooling}, P_{cooling\ load}) \quad (7)$$

System cooling power analysis for air-conditioner unit

We choose to model the energy saving by considering the case of retrofitting a traditional air-cooled vapor-compression air-conditioner unit with the radiative cooling device using water as the heat transfer fluid, as shown in Fig. R3a. The thermodynamic cycles are shown in Fig. R3b. The blue line represents the traditional air-cooled AC and the red line represents the AC coupled with radiatively-cooled, below-ambient-temperature water. Points 1-4 represent saturated vapor at low pressure, compressed vapor, saturated liquid at high pressure, liquid after expansion valve. The radiative cooling system provides additional cooling and reduces the condenser temperature, and the new thermodynamic cycle follows points 1'-4', which is more efficient than 1-4. By comparing

the coefficient of performance (COP) before and after installing the radiative cooler, the energy saving can be calculated.

Figure R3 Modeling cooling system-level energy savings. a, Scheme for air-cooled AC coupled with radiative-cooling fluid panels. **b**, Thermodynamic cycle diagram of the air-cooled AC with (red) and without (blue) radiative cooling.

For the air-cooled vapor-compression AC, the basic thermodynamic equations (eq.8 - eq.13) are listed below to demonstrate its performance during the “compression-condensation-expansion-evaporation” loop. P_{load} is the heat removed from the building for maintaining the comfortable room temperature (22°C) per unit time, which is calculated from *EnergyPlus*. dm/dt is the mass flow rate of refrigerant. h_1, h_2, h_3, h_4 are the enthalpies for 4 points in the thermodynamic cycle. P_r is the condenser heat rejection per unit time, which is also equal to the cooling power of the fan for the condenser. $h_a(v_{air})$ is the overall heat transfer coefficient of a finned tube heat exchanger, which is the function of air velocity. The formula of h_a is based on the empirical correlation of the Nusselt number of laminar flow on flat plates ((Frank P.; DeWitt, David P. (2007). *Fundamentals of heat and mass transfer* (6th ed.). Hoboken: Wiley.)) and the geometrical parameters of a common finned tube heat exchanger. T_{airin} and T_{airout} are the temperature for air flowing in and out of the condenser fins, respectively. P_{com} is the input power for the compressor. η_{com} is the compressor efficiency. $P_{fan}(v_{air})$ is the power consumption of the fan calculated by the fan affinity law. Hence, the COP of the

traditional AC can be calculated by dividing P_{load} with P_{total} .

$$P_{\text{load}} = \frac{dm}{dt}(h_1 - h_4) \quad (8)$$

$$P_r = \frac{dm}{dt}(h_2 - h_3) \quad (9)$$

$$P_r = h_a(v_{\text{air}}) \cdot (T_{\text{airout}} - T_{\text{airin}}) \quad (10)$$

$$P_{\text{com}} = \frac{dm}{dt}(h_2 - h_1) / \eta_{\text{com}} \quad (11)$$

$$P_{\text{total}} = P_{\text{fan}}(v_{\text{air}}) + P_{\text{com}} \quad (12)$$

$$\text{COP} = \frac{P_{\text{load}}}{P_{\text{total}}} \quad (13)$$

Then we added radiatively-cooled water panel to the system to enhance the efficiency. For the cooling panels, thermodynamic equations (eq.14-eq.18) were given below. Note p_{cool} is cooling power density (W/m^2) and P_{cool} is the cooling power (W). To account for the negative correlation between cooling power density and sub-ambient temperature drop (ΔT_{cool}) due to the hemispherical ambient thermal radiance, the cooling power density is subtracted by $4.23 \cdot \Delta T_{\text{cool}}$, which is based on Eriksson, T. S., and C. G. Granqvist's research (Eriksson, T. S., and C. G. Granqvist. "Radiative cooling computed for model atmospheres." *Applied Optics*, **21**, 4381-4388 (1982)). S is the effective roof area that could be utilized, which is assumed to be 60% of the model building rooftop area. dm_{water}/dt is the water mass flow rate inside the tube. C_{water} is the heat capacity of water. $h_w(v_{\text{water}})$ is the overall heat transfer coefficient of water approximated with the Dittus-Boelter equation for pipe flow. T_{waterin} and T_{waterout} are the temperature for water flowing in and out of the radiative cooling surface-plate heat exchanger, respectively. $P_{\text{pump}}(v_{\text{water}})$ is input water pump power, which is also calculated by the fan affinity law. The new COP could, therefore, be calculated by dividing P_{load} by P'_{total} .

$$(p_{\text{cool}} - 4.23 \cdot \Delta T_{\text{cool}})S = P_{\text{cool}} = \frac{dm_{\text{water}}}{dt} C_{\text{water}} \Delta T_{\text{cool}} \quad (14)$$

$$P'_r = h_a(v_{\text{air}}) \cdot (T_{\text{airout}} - T_{\text{airin}}) + h_w(v_{\text{water}}) \cdot (T_{\text{waterout}} - T_{\text{waterin}}) \quad (15)$$

$$P'_{\text{total}} = P_{\text{fan}} (v_{\text{air}}) + P_{\text{pump}} (v_{\text{water}}) + P_{\text{com}} (\Delta t_{\text{cool}}) \quad (16)$$

$$\text{COP}_{\text{new}} = \frac{P_{\text{load}}}{P'_{\text{total}}} \quad (17)$$

Finally, the cooling energy saved by using cooling materials can be demonstrated by:

$$E_{\text{saving,cool}} = E \left(1 - \frac{\text{COP}}{\text{COP}_{\text{new}}} \right) \quad (18)$$

where E is the cooling electricity consumption with traditional AC, calculated by *EnergyPlus*

Heating-mode energy saving:

For heating energy saving, eq.19 and 20 were used to analyze the device performance. P_{heating} is the radiative heating power of the device. I is the global horizontal solar radiation obtained from TMY3 weather data. S is the effective roof area that could be utilized, which is assumed to be 60% of the model building rooftop area. α is absorption coefficient of heating materials. $E_{\text{saving,heat}}$ is the heating energy saving.

$$P_{\text{heating}} = IS\alpha \quad (19)$$

$$E_{\text{saving,heat}} = 3600 * \text{MIN} (P_{\text{heating}}, P_{\text{heating load}}) \quad (20)$$

Through the above cooling model and heating model, the cooling and heating energy saving of each city per hour can be calculated. In dual-mode calculation, we can choose to operate in the mode that generates the maximum energy saving in that specific hour. That is, if the cooling saving is larger than heating saving, then we use cooling mode in this hour. Otherwise, heating mode would be taken into consideration.

$$E_{\text{saving,dual}} = \text{MAX} (E_{\text{saving,cool}}, E_{\text{saving,heat}}) \quad (21)$$

Therefore, by arranging all cooling and heating energy saving in each hour of each city, we could get the annul energy saving in the U.S. in heating-only, cooling-only, and dual-mode approaches.

This can be found in Page 11-16 of the revised Supporting information.

5. The required voltage to induce the high-thermal contact is in kV level, which seems too high for civil applications. Besides, the duration after charge removal in fig 2d is too short for an HVAC application that usually last for months. Therefore, I have major concern on the practicability at this point.

Our reply:

We appreciate the reviewer's comments. Even though the voltage looks high, the current is only about 0.07 mA, which is a safe current for the human body (David W. Smith, Preventing Electrical Shock, The Texas A&M University system, https://cdn-ext.agnet.tamu.edu/wp-content/uploads/2019/06/E-221_-_Preventing-Electrical-Shock.pdf). In addition, the working voltage of ionizer air purifiers, a common household appliance, can also reach 5 kV without the concern of electrical shock (<http://www.air-purifier-power.com/ionizer-air-purifier.html>). Accordingly, as long as proper management and control are implemented, we believe our high-voltage, low-power device would be safe for civil applications.

We have added it in Page 8 of the revised manuscript.

According to reviewer's suggestions, we have tested the electrostatic effect lifetime for one week. The results can be seen in revised Fig. 2d (see Fig. R4 below as well). While this static electricity may not last for several months, we believe this concern can be solved by periodic "charging" in practical applications. The time for each "charging" is completed within only a few seconds.

We have added it in Page 8 of the revised manuscript.

Fig. R4. Reducing the average thermal contact resistance by electrostatic effect. **a**, Schematic of dual-mode material morphology evolution as the function of surface static charges. As the static charge increases by either triboelectricity or applied voltage, the Maxwell pressure can increase both the macroscopic contact area and the local contact conductance, which significantly decreases the overall thermal contact resistance. **b**, Thermography images of the cooling material on a constant temperature substrate after applying 0, 0.5, 1, 2, and 3 kV, respectively. **c**, Average thermal conductance over applied voltage. **d**, The average thermal conductance remains unchanged for 3 days even after the applied voltage is removed because of the strong tendency of the PI film to retain surface charges.

6. The discussion in fig. 3c is insufficient. Although the authors try to explain the figure by referring fig. S7, it is still unclear regarding how the morphology or grain size affect the optical feature of the heating material.

Our reply:

We thank the reviewers for pointing this out. We have added more discussion about heating material in revised Note 2 (see below).

For heating material, on the copper film, a layer of zinc film of 1 μm thick was electrodeposited (voltage: 2 V, anode: zinc metal, electrolyte: 0.25 M $\text{ZnSO}_{4(\text{aq})}$), followed by galvanic replacement reaction with 0.2 M $\text{CuSO}_{4(\text{aq})}$, and the heating material was obtained after deionized water washing and drying. As shown in Fig. R5, it can be found with the increase of reaction time with $\text{CuSO}_{4(\text{aq})}$, the absorption of 300 - 2000 nm is increased while the absorption of 4 - 18 μm is decreased. The observations can be attributed to the size of copper/copper oxide clusters is increased (as shown in Fig. R6). Specifically, the absorption of 300 - 2000 nm stems from the localized surface plasmon resonances of the Cu nanoparticles. As the size of clusters increases, both near-field coupling and the volume of light-matter interactions increase, which promotes broadband absorption in the solar spectrum. For 4 - 18 μm part, the nanoparticle layer behaves as a lossy effective medium because of the small cluster size compared to thermal radiation wavelength. Therefore, longer reaction time leads to a higher attenuation of light in both solar and mid-IR regimes. (Mandal, J. et al. Scalable, “Dip-and-Dry” Fabrication of a Wide-Angle Plasmonic Selective Absorber for High-Efficiency Solar-Thermal Energy Conversion. *Advanced Materials* **29**, 1702156, (2017).).

Figure R5. Optical properties of heating materials. a and b, Reflectance spectra of heating materials of different reaction time with CuSO_4 .

Figure R6. SEM images of the heating materials. a. reaction time of 15 s. **b.** reaction time of 30 s. **c.** reaction time of 60 s.

We have added it in Page 8-9 in supporting information

Finally, the authors are very grateful to you for investing your valuable time in reviewing our manuscript again

Response to reviewer #2

The authors propose a dual-mode device and demonstrate that it is capable of switching between radiative cooling and solar heating for year-long energy savings both in the heating and cooling conditions. Designed actuators will physically rotate the cooling/heating films along a track system to switch between two different films. The work focuses on reducing the interfacial thermal resistance between the functioning layer and the substrate surface by applying a high voltage of 2 kV. The overall energy savings are estimated and compared with heating-only or cooling-only devices. A dual-mode device is beneficial to switch between the radiative cooling mode or solar heating mode at different seasons. However, we believe that the work is an incremental advance and does not have the novelty that is suitable for publication in Nature Communications. In addition, there are several concerns that need to be addressed.

Our reply:

We thank the reviewer for his/her comments. Indeed, there were many pioneering works in the fields of nighttime radiative cooling and solar heating that laid the foundation of the present work. However, they are all static or quasi-dynamic devices (see Table R1), which cannot completely solve the dynamic heating and cooling demand problem effectively, especially in the daytime. We are grateful for the inspirations and the shared knowledge. Still, we believe our work contains a good amount of elements of novelty. To better convey our ideas, we summarize our key accomplishments as follows:

1. **This is the first experimental demonstration of the solar heating and daytime radiative cooling active dual-function building envelope.** We have conducted extensive literature research to support this statement, which is compiled in Supporting Information Table 4 (see Table R1 as well). First of all, despite its significant potential to enhance the building energy efficiency, the concept of switchable or dual-mode building rooftops are still in an early stage. Among these papers, none of them truly accomplished both solar heating and daytime radiative cooling. **This is because of one fundamental limit – solar heating and daytime radiative cooling have exactly the opposite required optical properties.** Based on this fundamental limit, it can be concluded that the materials for these two modes must be physically or chemically changed. For adapting to various complicated scenarios, such tuning should be active rather than passive. Therefore, from the materials

science point of view, there are only two feasible approaches: electrochromism and mechanochromism. This paper chooses mechanochromism for its compatibility with existing building technologies and thus a higher chance of creating real social impacts. It was not a trivial task, but we eventually solved it using the electrostatic effect, as described below.

2. **This is the first paper to achieve reversible thermal contact for mechanochromic heating/cooling.** We realized that, in practice, all the tuning of material's optical properties would be in vain if the heating/cooling energy cannot be transported to the buildings or the heat exchangers. In mechanochromism that relies on frequent movement or shape-changing to tune the optical properties, this means the thermal contact needs to be tunable as well. To the best of our knowledge, this is the first paper to employ the concept of reversible electrostatic forces to achieve large-scale thermal contact switching for net-zero buildings.
3. **To make a broader impact, we estimate the energy saving of dual-mode (daytime radiative cooling and solar heating) device for the entire U.S. climate zones.** We pointed out the potential need for dual-mode building envelope by noting the needs for both heating and cooling during one year (the revised manuscript Fig. 1 and Fig. R1 below). After experimentally demonstrating the dual-mode device, we then calculated the building energy consumption and saving for different climate zones and cities. The results not only support our assumption but provide a guideline for future employment of the dual-mode device. We envision these data will be an essential piece of information to encourage the readers of Nature Communications from a variety of disciplines to contribute to the field of energy-efficiency buildings and sustainability.

Fig. R1. a, Annual heating and cooling degree days of 16 U.S. cities that represent the 16 climate zones. **b**, Heating and cooling degree days over 12 months in Durham, NC, USA.

	Daytime subambient cooling	Daytime heating	Nighttime subambient cooling	Heating performance Absorption (A), Emissivity (E)	Cooling performance Reflectance (R), Emissivity (E)	Tuning method
Hu et al. ¹	×	√	√	300-2000 nm A = ~ 90%; 8-13 μm E = ~ 80%	300-2000 nm A = ~ 90%; 8-13 μm E = ~ 80%	—
Vall et al. ²	×	√	√	—	—	—
Hu et al. ³	×	√	√	—	—	—
Ono et al. ⁴	×	—	—	8-13 μm E = 5.4%	8-13 μm E = 63.6%	Thermal
Mandal et al. ⁵	×	√	√	300-2000 nm A = ~90%; 4-18 μm E = ~ 90%	300-2000 nm R = ~88%; 8-13 μm E = ~60%	Electrical
Our work	√	√	√	300-2000 nm A = 93.4%; 4-18 μm E = 14.2%	300-2000 nm R = 97.3%; 8-13 μm E = 94.1%	Mechanical

Table R1. Summary of the tunable technologies based on solar heating and radiation cooling reported in the literature.

References for Table R1

[1] Hu, M. et al. Field test and preliminary analysis of a combined diurnal solar heating and nocturnal radiative cooling system. *Applied Energy*, **179**, 899-908 (2016).

[2] Vall, S. Medrano, M. Solé, C. & Castell, A. Combined radiative cooling and solar thermal collection: experimental proof of concept. *Energies*, **13**, 893 (2020).

[3] Hu, M. et al. Performance assessment of a trifunctional system integrating solar PV, solar thermal, and radiative sky cooling. *Applied Energy*, **260**, 114167 (2020).

[4] Ono, M. Chen, K. Li, W. & Fan, S. Self-adaptive radiative cooling based on phase change materials. *Optics express*, **2**, A777-A787 (2018).

[5] Mandal, J. et al. $\text{Li}_4\text{Ti}_5\text{O}_{12}$: A visible-to-infrared broadband electrochromic material for optical and thermal management. *Advanced Functional Materials*, **28**, 1802180 (2018).

1. The device relies on 2 kV to pull down the cooling or heating layers on the substrate to reduce interfacial thermal resistance. Although the device does not require continuous voltage application, the authors should elaborate on an economical solution to safely generate the required 2 kV high voltage within the device. It is critical to include the energy/equipment cost to initiate the 2 kV voltage that pulls down the cooling/ heating film. Additionally, the authors should include the installation and maintenance cost as well.

Our reply:

Thank you very much for your constructive suggestion. Even though the voltage looks high, the current is only about 0.07 mA, which is a safe current for the human body (David W. Smith, Preventing Electrical Shock, The Texas A&M University system, https://cdn-ext.agnet.tamu.edu/wp-content/uploads/2019/06/E-221_-Preventing-Electrical-Shock.pdf). In addition, the working voltage of ionizer air purifiers, a common household appliance, can also reach 5 kV without the concern of electrical shock (<http://www.air-purifier-power.com/ionizer-air-purifier.html>). Accordingly, as long as proper management and control are implemented, we believe our high-voltage, low-power device would be safe for civil applications.

We have added it in Page 8 of the revised manuscript.

1.Economic analysis

The cost analysis for this study was based on the post-1980 medium office model defined by the U.S. Department of Energy. This model contains three stories with a roof area of 1660 m². The cover area of device is assumed to be 60%, which is equivalent to an area of 996 m². Here, we assume the device contains four major parts: two 4'' PVC hollow rod (31.56 m), two electric motors for switching the heating and cooling materials, a high-voltage electrostatic generator to enhance the thermal contact and two programmable logic modules to control the electrostatic generator and electric motors respectively. The total cost of the device is composed of the

equipment cost, installation fee, and energy consumption associated with each component. More details about that will be discussed below.

In this device, the PVC hollow rods are used as the rollers for heating and cooling materials. During the switching process, electric motors rotate the PVC hollow rod to change the heating mode or cooling mode. Based on the average normal life expectancy: electric motors (17.1 years) (<https://www.motionindustries.com/knowledgesites/motors/article.jsp?slug=what-is-the-normal-life-expectancy-of-a-motor> [Accessed May 28, 2020]), PVC hollow rod (40 years) (<https://santhoffplumbingco.com/the-lifespan-of-residential-plumbing-pipes/> [Accessed May 28, 2020]), electrostatic generator (estimated 5 year) and programmable logic controller (11.4 years) (<http://www.ti.com/lit/an/sprabv9a/sprabv9a.pdf?ts=1588469624577> [Accessed May 28, 2020]), as shown in Table R2, the 10-years cost is about \$608. The price evaluation of heating material and cooling material refers to similar products in the market (see Table R2).

Besides the equipment and material cost, labor fee is another aspect to consider for cost analysis. Based on the life expectancy in Table R2, the labor fee mainly comes from the initial assembling and annual change of electrostatic generator. As shown in Table R3, based on average labor rate for engineers (\$46.8/hour) from U.S. Bureau of Labor (<https://www.bls.gov/oes/2017/may/oes172199.htm> [Accessed May 28, 2020]), the 10-year cost is about \$281. Table R4 summarizes the energy consumption associated with the electric motors and the electrostatic generator. The total process time is assumed within 3 minutes (see Video S1). After the switching is complete, the electrostatic generator charges the film at 2 kV for about 1 minute to enhance the thermal contact. In the United States, the electricity rate ranges from 8.8 cent/kwh to 32.1 cent/kwh. Based on the electricity rate in North Carolina (11 cent/kwh) (<https://www.electricchoice.com/electricity-prices-by-state/> [Accessed May 28, 2020]), 10-year cost for dual mode is about \$8.3 (see table R4).

Compared with heating mode only and cooling mode only, the dual-mode cost \$8 (Electricity rate) + \$281 (Labor fee) + \$608 (Equipment cost) + \$857 (Material cost) = \$1754 more in 10 years (Cooling-only and heating-only do not need installation of motors, electrostatic generators, and logic modules.).

Table R2. Equipment and material cost of the dual-model device.

Equipment	Units	Unit cost (\$)	Cost	Lifetime (years)	10 years cost
PVC hollow rod (GRAINGER, Nominal 4", Length 10 ft)	11	\$55	\$605	40	\$151
Electric Motors (ZOZHI, 3 phase 1 hp 750 watt ac gear motor 900 rpm 380V MS90S-6)	2	\$45.5	\$91	17	\$53
Stainless Steel Wall Pipe Support (Ventis, VDV-WS04)	4	\$26	\$104	10	\$104
Electrostatic Generator (ebay, 12V NEW Adjustable High-voltage Electrostatic Generator / Negative Ion Generator)	1	\$15	\$15	5	\$30
Logic module (Siemens, LOGO! 24 CE)	2	\$154	\$308	11.4	\$270
Heating material (Refer to Cooling material)	-	\$0.86/m ²	\$857	~10	\$857
Cooling material (Refer to Highly Reflective Mylar Film, Amazon)	-	\$0.86/m ²	\$857	~10	\$857

Total cost					\$ 2322
------------	--	--	--	--	---------

Table R3. Labor cost associated with each component in the building envelope.

Equipment	Labor Unit	Hour	Labor rate (\$/hour)	10 years cost
Initial logic module programing	1	2	46.8	\$93.6
Maintaining	1	2	46.8	\$93.6
12 V High-voltage electrostatic generator changing	1	2	46.8	\$93.6
Total cost				\$281

Table R4. Energy consumption associated with electric motors and electrostatic generator.

Equipment	Power (W)	Operation Time (s)	Annual Frequency	Annual Energy (kWh)	10 years cost
High-voltage Electrostatic Generator	5	60	180	0.015	\$1.65e-2
Electric motor (Two)	750	180	100	7.5	\$8.3
Total electricity cost					\$8.3

2. The work does not address some of the reliability issues when dirt or water gets into the interface between the PI film and substrate, or inside the track system. If there is dirt accumulating on the

substrate beneath the films under windy conditions, the high voltage may not be sufficient to reduce the interfacial thermal resistance. Furthermore, water or humidity is also known to reduce the effect of static electricity. These are especially important when the films are exposed to the ambient air.

Our reply:

Thank you very much for your constructive suggestion. The analysis of reliability issues is listed below.

Analysis of the potential impact of dirt

It is expected that the interfacial thermal resistance could be impacted when dirt gets into the interface between the PI film and the substrate. When manufacturing the proposed dual-mode heating and cooling device, it is critical to package the film, the underneath substrate, and the rolling components into a tightly sealed modular system to highly reduce the dirt penetration risk. In addition, operation maintenance architectures such as brush cleaners installed at the separation edge of the heating/cooling film will intermittently clean the surface of the underneath substrate and help remove dirt that may appear between PI film and substrate.

Analysis of the potential impact of humidity

As shown Fig. R2, the effect of humidity on interfacial thermal resistance is also demonstrated. The result shows that the good thermal contact can maintain for two days when the relative humidity is in the range of 40% to 60%. Even in the rare case of > 95% humidity, a good thermal contact can still last for more than one day. In practical applications, periodic “charging” will be performed to ensure continuous and good thermal contact.

Figure R2. The thermal contact over time at different humidity.

We have added it in Page 7 of the revised Supporting information.

Finally, the authors are very grateful to you for investing your valuable time in reviewing our manuscript again

Response to reviewer #3

This paper presented the technical apparatus and performance of a novel dual-mode radiant heating and cooling panel which can, most critically, reject heat to the ambient sky even during sunny daytime conditions. The authors provide a validation of experimental data, followed by calibration of an empirical model, followed by the use of this model (coupled with a building energy model) to assess the potential cooling and heating savings offered by the technology across the United States. My comments regarding the paper are provided in the following sections:

Our reply:

We thank the reviewer for his/her evaluation. The point-by-point response on the detailed comments is as follows.

(1) About the Physical apparatus

- I do not pose sufficient expertise in materials engineering to understand and critique the methods, results and discussion surrounding the physical performance of individual panels. I trust this will be captured by other reviewers. However, I do not perceive the technical description and analysis of the panel itself to be the core proposed novelty of the paper, nor the means by which the authors demonstrate the proposed impact of their innovation.

Our reply:

We thank the reviewer for his/her comments. Indeed, there were many pioneering works in the fields of nighttime radiative cooling and solar heating that laid the foundation of the present work. However, they are all static or quasi-dynamic devices (see Table R1), which cannot completely solve the dynamic heating and cooling demand problem effectively, especially in the daytime. We are grateful for the inspirations and the shared knowledge. Still, we believe our work contains a good amount of elements of novelty. To better convey our ideas, we summarize our key accomplishments as follows:

1. **This is the first experimental demonstration of the solar heating and daytime radiative cooling dual-function active building envelope.** We have conducted extensive literature research to support this statement, which is compiled in Supporting Information Table 4 (see Table R1 as well). First of all, despite its significant potential to enhance the

building energy efficiency, the concept of switchable or dual-mode building rooftops are still in an early stage. Among these papers, none of them truly accomplished both solar heating and daytime radiative cooling. **This is because of one fundamental limit – solar heating and daytime radiative cooling have exactly the opposite required optical properties.** Based on this fundamental limit, it can be concluded that the materials for these two modes must be physically or chemically changed. For adapting to various complicated scenarios, such tuning should be active rather than passive. Therefore, from the materials science point of view, there are only two feasible approaches: electrochromism and mechanochromism. This paper chooses mechanochromism for its compatibility with existing building technologies and thus a higher chance of creating real social impacts. It was not a trivial task, but we eventually solved it using the electrostatic effect, as described below.

2. **This is the first paper to achieve reversible thermal contact for mechanochromic heating/cooling.** We realized that, in practice, all the tuning of material's optical properties would be in vain if the heating/cooling energy cannot be transported to the buildings or the heat exchangers. In mechanochromism that relies on frequent movement or shape-changing to tune the optical properties, this means the thermal contact needs to be tunable as well. To the best of our knowledge, this is the first paper to employ the concept of reversible electrostatic forces to achieve large-scale thermal contact switching for net-zero buildings.
3. **To make a broader impact, we estimate the energy saving of dual-mode (daytime radiative cooling and solar heating) device for the entire U.S. climate zones.** We pointed out the potential need for dual-mode building envelope by noting the needs for both heating and cooling during one year (the revised manuscript Fig. 1 and Fig. R1 below). After experimentally demonstrating the dual-mode device, we then calculated the building energy consumption and saving for different climate zones and cities. The results not only support our assumption but provide a guideline for future employment of the dual-mode device. Admittedly, the calculation process can be further improved in the future. Still, we aimed to present these data to encourage the readers of Nature Communications from a variety of disciplines to contribute to the field of energy-efficiency buildings and sustainability.

Table R1. Summary of the synergistic technologies based on solar heating and radiation cooling reported in literature.

	Daytime subambient cooling	Daytime heating	Nighttime subambient cooling	Heating performance Absorption (A), Emissivity (E)	Cooling performance Reflectance (R), Emissivity (E)	Tuning method
Hu et al. ¹	×	√	√	300-2000 nm A = ~ 90%; 8-13 μm E = ~ 80%	300-2000 nm A = ~ 90%; 8-13 μm E = ~ 80%	—
Vall et al. ²	×	√	√	—	—	—
Hu et al. ³	×	√	√	—	—	—
Ono et al. ⁴	×	—	—	8-13 μm E = 5.4%	8-13 μm E = 63.6%	Thermal
Mandal et al. ⁵	×	√	√	300-2000 nm A = ~ 90%; 4-18 μm E = ~ 90%	300-2000 nm R = ~88%; 8-13 μm E = ~60%	Electrical
Our work	√	√	√	300-2000 nm A = 93.4%; 4-18 μm E = 14.2%	300-2000 nm R = 97.3%; 8-13 μm E = 94.1%	Mechanical

References for Table R1.

[1] Hu, M. et al. Field test and preliminary analysis of a combined diurnal solar heating and nocturnal radiative cooling system. *Applied Energy*, **179**, 899-908 (2016).

- [2] Vall, S. Medrano, M. Solé, C. & Castell, A. Combined radiative cooling and solar thermal collection: experimental proof of concept. *Energies*, **13**, 893 (2020).
- [3] Hu, M. et al. Performance assessment of a trifunctional system integrating solar PV, solar thermal, and radiative sky cooling. *Applied Energy*, **260**, 114167 (2020).
- [4] Ono, M. Chen, K. Li, W. & Fan, S. Self-adaptive radiative cooling based on phase change materials. *Optics express*, **2**, A777-A787 (2018).
- [5] Mandal, J. et al. $\text{Li}_4\text{Ti}_5\text{O}_{12}$: A visible-to-infrared broadband electrochromic material for optical and thermal management. *Advanced Functional Materials*, **28**, 1802180 (2018).

(2) About the climate analysis undertaken

- I feel the main overarching disadvantage of the manuscript is an apparent oversimplification, or misrepresentation, of data and results that aim to assess the viability of the proposed technology for heating and cooling across the US. This begins early in the manuscript in the presentation of a classification of cities with regards to 'heat management monotony'. I'm not familiar with such a term, but the analysis summarised in Figure 1a is counter-intuitive and seems to disagree with supplemental data provided by the authors in Figure S4. For example, from Figure 1a, Seattle appears to be a more cooling dominant city than San Francisco, but clearly from Figure S4, San Francisco very clearly possesses a larger and more persistent cooling load than Seattle, and Seattle conversely possesses a much larger heating load. The authors should consider reverting to more widely-understood metrics for heating/cooling classification of cities such as Heating Degree Days and Cooling Degree Days.

Our reply:

We thank the reviewers for the suggestion. We have removed the previous expression and used the Heating Degree Days and Cooling Degree Days to describe the requirement of heating and cooling of building in the revised manuscript, Fig.1. (see Fig. R1 below as well).

Fig. R1. a, Annual heating and cooling degree days of 16 U.S. cities that represent the 16 climate zones. **b**, Heating and cooling degree days over 12 months in Durham, NC, USA.

We revised the manuscript accordingly as follows:

“...heating degree days and cooling degree days can commonly and quantitatively describe the heating and cooling demands of buildings⁵. Fig. 1a shows the annual heating and cooling degree days of 16 U.S. cities that represent the 16 climate zones. It can be found most cities need both heating and cooling in the whole year. Taking Durham, North Carolina as an example, the cooling consumption predominates from May to October, and the rest 6 months are heating-dominant (Fig. 1b).”

This can be found in page 2 of the manuscript.

(3) About the predictive model of radiant cooling

- The model assumes that the main intermediary between the panel and outer space is the atmosphere. This is true, but it would appear that the chosen model assumes a clear sky exists at all times and so transmissivity is only affected by the vertical water column. As it is the intent of this paper to examine the applicability of the radiant cooling panel across the continental US on an annual basis, this assumption cannot hold. Accounting for the effect of cloud cover is

challenging, but it is necessary to do so. Not only do clouds change the radiant interaction between the panel and the edge of the atmosphere, depending on their altitude and temperature, they may be a source of radiation in their own right.

Our reply:

Thank you very much for your constructive suggestion. According to Reviewer’s suggestions, we have considered the cloud cover and recalculated the model (see Fig. 5 in the revised manuscript, see Fig. R2 below as well). The more details see below.

Potentially, there are quite many application methods for the proposed dual-mode radiation heating and radiative cooling materials in buildings. We demonstrate an example application of integrating the material with building envelopes to provide space heating and cooling energy using heat exchangers. At this system level application, a comprehensive integrated analysis of the proposed device and the subject building is needed, which creates hourly performance simulation for the 16 cities throughout a whole year. In addition, in order to evaluate the energy savings of the dual-mode device under real application condition, the typical meteorological year (TMY3) (National Solar Radiation Data Base, 1991-2005 Update: Typical Meteorological Year 3; [https://rredc.nrel.gov/solar/old_data/nsrdb/1991-2005/tmy3/.](https://rredc.nrel.gov/solar/old_data/nsrdb/1991-2005/tmy3/)) weather data are used, and the impacts of the humidity and the clouds on cooling capability are evaluated. Therefore, the following calculation algorithms (eq.1 – eq.6) are selected to estimate the cooling power with effective atmospheric emissivity (ε_{atm}):

$$P_{cooling\ power} = P'_{rad} - P'_{atm} \tag{1}$$

$$P'_{rad} = A\varepsilon_{film}\sigma T_{film}^4 \tag{2}$$

$$P'_{atm} = A\varepsilon_{film}\varepsilon_{atm}\sigma T_{amb}^4 \tag{3}$$

where ε_{film} and T_{film} are the emissivity and surface temperature of the film, and A is the area. The ε_{atm} is given by⁵,

$$\varepsilon_{atm} = \varepsilon_{atm,c} (1-0.78CF) + 0.38CF^{0.95} RH^{0.17} \tag{4}$$

$$\varepsilon_{atm,c} = 0.618 + 0.056 \sqrt{P_w} \tag{5}$$

$$P_w = P_0 \exp [(c_T T_d) / (T_d + T_0)] \quad (6)$$

Where $\varepsilon_{atm,c}$ is the effective sky emissivity under clear skies, CF is the cloud fraction, RH is the ambient relative humidity, and P_w is the ambient water vapor partial pressure, T_d is the dew point, $P_0 = 610.94$ Pa, $c_T = 17.625$, and $T_0 = 243.04$ °C. The hourly values of these weather-related parameters can be obtained from TMY3 weather data.

This can be found in Page 11 of the revised Supporting information.

Fig. R2 Modeled annual energy saving of different energy-saving modes. Building energy saving map for all U.S. climate zones with **a**, only heating mode, **b**, only cooling mode, and **c**, dual-mode, **d**, annual averaged energy saving at different modes and energy loads over different cities. The calculation indicates that the dual-mode device can save 236 GJ (19.2% of the heating and cooling energy), which is 1.7 times higher than cooling-only (138 GJ) and 2.2 times higher than heating-only (106 GJ) devices.

(4) Spatial obstructions and view factors

- The model assumes that the effective roof area that can be covered by the proposed radiant panel system may be 60%. No further information is given regarding the effective sky view factor of rooftop systems. This surely cannot be 1 due to the diversity of likely obstructions that typical rooftops face, from surrounding vegetation to surrounding buildings.

Our reply:

We appreciate the reviewer's comments. Here, we assume the sky view factor is one based on two reasons. First, our simulation aimed to provide a prospect for dual-mode application on a large scale. Accounting for the real effect of sky view factor requires case-by-case analysis of buildings and the surrounding landscape, which can be an excellent research project but also, in our opinion, beyond the scope of this manuscript. Second, even when there are obstructions around the building, the sky view factor can still be largely maintained by geometric optical design such as tapered guides (Zhou, L. et al. A polydimethylsiloxane-coated metal structure for all-day radiative cooling. *Nature Sustainability* **2**, 718-724, (2019)).

(5) Other viable interactions between the rooftop and the sun.

- During summer daytime periods, it would appear that there is still an argument to prefer utilisation of roof area for solar PV generation. Electricity generation can serve multiple uses, from building to (increasingly) transport energy services. Modern solar PV systems may generate well over 150 W/m² in electricity. Under the same form of system modelling as applied in this work, one could envisage this solar PV electricity serving a heat pump-cooling system with a COP in excess of 3. From an energy consumption mitigation standpoint, this would seem to be a far more effective solution for reducing current electricity usage for space cooling across the United States. On the other hand, an understated advantage of the proposed solution is that it can operate under its 'cooling' mode during both daytime and nighttime. This does pose questions, however. Would the system operate at times of night where an immediate indoor cooling demand does not exist, and therefore it would be valuable to store chilled water? If so, how would such a system perform, particularly with respect to exergy? In general, the underdeveloped nature of the energy model, from unclear assumptions made regarding building typologies (is an 1660 m² office building meant

to be representative of the US building stock?), to unclear interactions between the radiant cooling panel and dynamic weather as discussed, would suggest more thoughtful analysis or explanation of this part of the provided work is needed. Ultimately, I would propose that, if building energy simulation is to be relied on to forecast the viability of the proposed radiant heating / cooling system to entire building stocks, greater consultation of prior research from the building simulation is needed, as is adoption of existing best practises with regards to thermal energy systems modelling.

Our reply:

We appreciate the reviewer's comments. We think the relationship between our devices and solar cells is not a competitive relationship, but a compatible relationship. For example, a project we are working on is to combine heating material, cooling material, and solar cells to achieve the triple-mode system (see Fig. R3) to more efficiently meet the different needs of buildings for heating, cooling and electricity under different situations, such as seasonal fluctuation, climate zone dependence, preference of residents, etc.

Fig. R3. Schematic of triple-mode (Cooling mode, Electricity mode and Heating mode) system. It can achieve cooling mode when cooling material switch on solar cell, electricity mode when transparent film is used, and heating mode when heating material is used.

Here, in our system, we assume that, if there is no need for cooling, the device will maintain at the photovoltaic mode. Of course, if we can use some possible technology to store the chilled water, we may get higher efficiency. We think this would be a new direction worthy of further exploration based on the current manuscript. As mentioned above, our simulation aimed to provide a prospect for dual-mode application on a large scale. So, we selected a building model that has been used many times in similar calculations (Goldstein, E. A., Raman, A. P. & Fan, S. Sub-ambient non-evaporative

fluid cooling with the sky. *Nature Energy* **2**, 17143, (2017); Li, T. Zhai, Y. He, S. Gan, W. Wei, Z. Heidarinejad, M. Dalgo, D. Mi, R. Zhao, X. Song, J. Dai, J. Chen, C. Aili, A. Vellore, A. Martini, A. Yang, R. Srebric, J. Yin, X. Hu, L. A radiative cooling structural material. *Science* **364**, 760-763 (2019).) for calculation.

Finally, the authors are very grateful to you for investing your valuable time in reviewing our manuscript again.

REVIEWER COMMENTS

Reviewer #1 (Remarks to the Author):

In this response, I appreciate the authors' major efforts to address most of my concerns. However, I still have two comments before my final recommendation.

1. In their response to my 6th comments, the authors added more detailed discussion about the heating material. In this section, they claimed that "As the size of clusters increases, both near-field coupling and the volume of light-matter interactions increase, which promotes broadband absorption in the solar spectrum." I agree that a strong absorption peak can be obtained by light coupling of nanoparticles. However, the increased cluster size only affects the wavelength of resonance. I do not see how the size of cluster promotes broadband absorption.

In addition, in the cited article at the end of this section (i.e. *Advanced Materials* 29, 1702156, (2017)), the explanation for broadband absorption was stated as "A broader absorption (i.e., lowered reflectance) extending into the infrared wavelengths due to resonance peaks is also seen with increasing d , (Figure 3b) although simulations indicate that h has a stronger effect." One can clearly see from its Fig. 3b that the difference size (d) of cluster does not significantly change the absorption bandwidth. Instead, the thickness (h) broadens the absorption. Therefore, I do not think the discussion is convincing in this revised section. Clarification at this point is required.

2. As though I do agree the novelty of this work using an electrostatically-controlled thermal contact for switchable thermal regulation, it is NOT the first demonstration of the dual functioning design for solar heating and radiative cooling, which is stated in the authors' response to reviewer 2. In 2019, an article published by Mandal et. al. already demonstrated a dynamic switching from opaque to transparent by wetting a porous film with refractive index matched fluid (doi.org/10.1016/j.joule.2019.09.016). In a recent work published by *Advanced Material* (DOI: 10.1002/adma.202000870), by simply stretching/compressing a porous PDMS film, the material can also switch between opaque and transparent state, hence obtaining mode switching when incorporated with a black back coating. Thereby, the authors need to better clarify the innovation of the work.

Reviewer #2 (Remarks to the Author):

The paper used a mechanical system to switch between radiative cooling film and absorbing film to alternate between cooling and heating mode. Static voltage of 2kV is used to ensure a good thermal contact of the film to the substrate. In this revision, the authors addressed the concern about the safety of the device, the reliability with dirt, and elaborated more details of the field test device. The revision addressed all of our concerns. We would like to recommend the publication.

Reviewer #1

In this response, I appreciate the authors' major efforts to address most of my concerns. However, I still have two comments before my final recommendation.

Our reply:

We thank the reviewer very much for his/her positive report and constructive suggestions. In the following sections, we would address the questions point-by-point.

1. In their response to my 6th comments, the authors added more detailed discussion about the heating material. In this section, they claimed that “As the size of clusters increases, both near-field coupling and the volume of light–matter interactions increase, which promotes broadband absorption in the solar spectrum.” I agree that a strong absorption peak can be obtained by light coupling of nanoparticles. However, the increased cluster size only affects the wavelength of resonance. I do not see how the size of cluster promotes broadband absorption.

In addition, in the cited article at the end of this section (i.e. *Advanced Materials* 29, 1702156, (2017)), the explanation for broadband absorption was stated as “A broader absorption (i.e., lowered reflectance) extending into the infrared wavelengths due to resonance peaks is also seen with increasing d , (Figure 3b) although simulations indicate that h has a stronger effect.” One can clearly see from its Fig. 3b that the difference size (d) of cluster does not significantly change the absorption bandwidth. Instead, the thickness (h) broadens the absorption. Therefore, I do not think the discussion is convincing in this revised section. Clarification at this point is required.

Our reply:

Thank you very much for your professional suggestion. As shown in Figure R1, it can be seen that the particle cluster size has a wide distribution, which is the basis of broadband optical absorption. As the reaction time increased, the average size of the particle became larger, and the distribution remained highly diversified, which again resulted in broadband absorption rather than the red-shift behavior. The increase of absorption intensity is explained by the increased total volume of particles, which is corroborated by the elemental analysis showing that Cu content increased from 16wt% to 21wt as the reaction time increased. Therefore, as the reaction time increases, a broadband increase of absorption occurs. In other words, we agree with the reviewer's comment and prediction that larger cluster particle size would result in the resonance wavelength shift, but such phenomenon did not appear in our experiment due to the broad size distribution, which was not captured by optical simulation and theory in the cited article (*Advanced Materials* 29, 1702156, (2017)).

Figure R1. The scanning electron microscope and Cu element mapping images of heating materials with the reaction time of (a,b) 15 s, (c,d) 30 s, and (e,f) 60 s.

To make these points clear, we have included the new SEM and EDX elemental mapping (Figure R1) in Figure S5 and corrected the description in revised supporting information, page 8. We also found and fixed a typo in the previous version regarding the dependence of mid-IR absorption on reaction time. We copy the section below for the reviewer's reference.

“For heating material, on the copper film, a layer of zinc film of 1 μm thick was electrodeposited (voltage: 2 V, anode: zinc metal, electrolyte: 0.25 M $\text{ZnSO}_{4(\text{aq})}$), followed by galvanic replacement reaction with 0.12 mM $\text{CuSO}_{4(\text{aq})}$, and the heating material was obtained after deionized water washing and drying. As shown in Fig. S4, it can be found with the increase of reaction time with $\text{CuSO}_{4(\text{aq})}$, **the absorption of both 300 - 2000 nm and 4 - 18 μm increased**. The observations can be attributed to the size of copper/copper oxide clusters is increased (as shown in Fig. S5). Specifically, the absorption of 300 - 2000 nm stems from the localized surface plasmon resonances of the Cu nanoparticles. **The wide size distribution of the Cu nanoparticle clusters results in broadband absorption, which is beneficial for solar heating. As the reaction time increases, both near-field coupling and the total volume of light-matter interactions increase, which promotes broadband absorption in the solar spectrum.** For 4 - 18 μm part, the nanoparticle layer behaves as a lossy effective medium because of the small cluster size compared to thermal radiation wavelength. Therefore, longer reaction time leads to a higher attenuation of light in both solar and mid-IR regimes³.”

2. As though I do agree the novelty of this work using an electrostatically-controlled thermal contact for switchable thermal regulation, it is NOT the first demonstration of the dual

functioning design for solar heating and radiative cooling, which is stated in the authors' response to reviewer 2. In 2019, an article published by Mandal et. al. already demonstrated a dynamic switching from opaque to transparent by wetting a porous film with refractive index matched fluid (doi.org/10.1016/j.joule.2019.09.016). In a recent work published by Advanced Material (DOI: [10.1002/adma.202000870](https://doi.org/10.1002/adma.202000870)), by simply stretching/compressing a porous PDMS film, the material can also switch between opaque and transparent state, hence obtaining mode switching when incorporated with a black back coating. Thereby, the authors need to better clarify the innovation of the work.

Our reply:

We thank the reviewers for pointing this out. After carefully comparing the literature and our result, we believe a more appropriate statement is that our device is the first experimental demonstration of the solar heating with selective absorber and daytime radiative cooling active dual-function building envelope (see Table R1).

Table R1. Summary and comparison of the synergistic technologies based on solar heating and radiative cooling reported in literature.

	Daytime sub-ambient cooling	Solar heating with selective absorber	Heating performance Absorption (A) Emissivity (E) Transmittance (T)	Cooling performance Reflectance (R) Emissivity (E)	Tuning method
Hu et al. ¹	×	×	300-2000 nm A = ~ 90%; 8-13 μm E = ~ 80%	300-2000 nm A = ~ 90%; 8-13 μm E = ~ 80%	—
Vall et al. ²	×	✓	Commercial Ti absorber	Black paint	(Hypothetical) Mechanical
Hu et al. ³	×	×	—	—	Tandem approach
Ono et al. ⁴	✓	×	8-13 μm E = 5.4%	8-13 μm E = 63.6%	Thermal
Mandal et al. ⁵	×	×	300-2000 nm A = ~ 90%; 4-18 μm E = ~ 90%	300-2000 nm R = ~88%; 8-13 μm E = ~60%	Electrical
Mandal et al. ⁶	✓	×	300-2000 nm T = ~ 94%; 4-18 μm E > 90% (Water)	300-2000 nm R = ~95%; 8-13 μm	Mechanical
Zhao et al. ⁷	✓	×	300-2000 nm A = ~ 95%; 4-18 μm E = ~ 94% (PDMS)	300-2000 nm R = ~93%; 8-13 μm E = ~94%	Mechanical
Our work	✓	✓	300-2000 nm A = 93.4%; 4-18 μm E = 14.2%	300-2000 nm R = 97.3%; 8-13 μm E = 94.1%	Mechanical

- [1] Hu, M. et al. Field test and preliminary analysis of a combined diurnal solar heating and nocturnal radiative cooling system. *Applied Energy*, **179**, 899-908 (2016).
- [2] Vall, S. Medrano, M. Solé, C. & Castell, A. Combined radiative cooling and solar thermal collection: experimental proof of concept. *Energies*, **13**, 893 (2020).
- [3] Hu, M. et al. Performance assessment of a trifunctional system integrating solar PV, solar thermal, and radiative sky cooling. *Applied Energy*, **260**, 114167 (2020).
- [4] Ono, M. Chen, K. Li, W. & Fan, S. Self-adaptive radiative cooling based on phase change materials. *Optics express*, **2**, A777-A787 (2018).
- [5] Mandal, J. et al. Li₄Ti₅O₁₂: A visible-to-infrared broadband electrochromic material for optical and thermal management. *Advanced Functional Materials*, **28**, 1802180 (2018).
- [6] Mandal, J. et al. Porous polymers with switchable optical transmittance for optical and thermal regulation. *Joule*, **3**, 3088-3099 (2019).
- [7] Zhao, et al. Switchable cavitation in silicone coatings for energy-saving cooling and heating. *Adv. Mater*, **32**, 2000870 (2020).

Indeed, the two articles mentioned by the reviewer (doi.org/10.1016/j.joule.2019.09.016 and DOI: 10.1002/adma.202000870) demonstrated tunable heating and cooling, but neither of them achieves low-emissivity in the solar heating mode (in other words, they are not selective absorber). The article published by Mandal et al. uses solvent in the solar heating mode, and the article published by Zhao et al. uses PDMS, both of which are high emissivity materials. The high emissivity poses a fundamental upper limit of their heating performance. Using the Stefan-Boltzmann's law and a simplified atmospheric window approximation, we used Eqn (1) to calculate the high-emissivity radiative heat loss of the two articles can be more than 160 W/m² higher than our selective absorber (Figure R2).

$$J = \varepsilon \cdot \sigma \cdot (T_s^4 - T_{amb}^4) \cdot (1 - p) + \varepsilon \cdot \sigma \cdot (T_s^4 - T_{uni}^4) \cdot p \quad (1)$$

where J is radiative power density, ε is emissivity of heating material, σ is the Stefan-Boltzmann constant, T_s , T_{amb} and T_{uni} are the temperature of heating material, ambient and deep space, respectively. p is the ratio of 8-13 μ m atmospheric window to the entire black body radiation spectrum. In sum, while these prior reports achieved dual-functionality both in an elegant and interesting manner, the choice of high emissivity material is not ideal for solar heating and therefore is fundamentally different from our work that uses the selective absorber.

Figure R2. Radiative power density over the emissivity of heating material. Here we assume the temperature of solar heating material and ambient are 303 K and 288 K. The transmittance of air windows wavelength (8-13 μm) is 1.

To make these points clear, we have corrected the description in the revised manuscript (page 3) and supporting information (page 21) (see below as well).

“To accomplish this goal, we resort to two infinite radiative heat source and cold source: the Sun (5800 K) and the outer space (3 K), respectively, to supply heating and cooling to buildings without using fossil fuels. For ideal daytime radiative cooling materials, the materials should have high reflectance in 200 - 2500 nm and high emissivity in 8-13 μm . For ideal solar heating, it is expected the material has high absorption in 200 – 2500 nm and low emissivity in > 2500 nm. Prior research efforts for both solar heating⁶⁻⁹ and radiative cooling¹⁰⁻²⁸ have yielded both high technological performance and deep scientific understanding, which spans from a variety of fields, including materials science, photonics and plasmonics, and heat transfer. However, they are mostly static or quasi-dynamic devices, which cannot completely solve the dynamic heating and cooling demand problem effectively, especially in the daytime. For the few prior reports of dynamic solar and mid-IR heat management, none of them demonstrate the ideal properties for both modes – selective absorber for solar heating and highly solar reflective layer for daytime radiative cool

cooling (Table S1). In other words, the heating/cooling performance was sacrificed for the dynamic tunability. Here, we demonstrate the dual-mode smart heat managing device that possess the ideal dual-mode optical properties and can achieve up to 71.6 W/m² of cooling power density and up to 643.4 W/m² of heating power density (over 93% of solar energy can be utilized) from experimental tests by optimizing the optical, mechanical, and heat transfer properties at various scales, ranging from nanoscale surface morphology to device-level design. We also performed the rigorous calculation of building energy efficiency that encompasses most of major cities in the U.S. to create the energy saving map for various climate zones. The map shows our dual-mode device outperforms the solar-heating-only and radiative-cooling-only devices, which can save 19.2% of building heating and cooling energy on average.”

Finally, the authors are very grateful to you for investing your valuable time to help us to improve the quality of our manuscript again.